# The formation of the Indo-Pacific montane avifauna

Andrew Hart Reeve [1] ✉, Jonathan David Kennedy[1], José Martín Pujolar[1,2], Bent Petersen [3,4], Mozes P. K. Blom[5], Per Alström [6], Tri Haryoko [7], Per G. P. Ericson [8], Martin Irestedt [8], Johan A. A. Nylander [8] & Knud Andreas Jønsson [1,8]

The processes generating the earth's montane biodiversity remain a matter of debate. Two contrasting hypotheses have been advanced to explain how montane populations form: via direct colonization from other mountains, or, alternatively, via upslope range shifts from adjacent lowland areas. We seek to reconcile these apparently conflicting hypotheses by asking whether a species' ancestral geographic origin determines its mode of mountain colonization. Island-dwelling passerine birds at the faunal crossroads between Eurasia and Australo-Papua provide an ideal study system. We recover the phylogenetic relationships of the region's montane species and reconstruct their ancestral geographic ranges, elevational ranges, and migratory behavior. We also perform genomic population studies of three super-dispersive montane species/ clades with broad island distributions. Eurasian-origin species populated archipelagos via direct colonization between mountains. This mode of colonization appears related to ancestral adaptations to cold and seasonal climates, specifically short-distance migration. Australo-Papuan-origin mountain populations, by contrast, evolved from lowland ancestors, and highland distribution mostly precludes their further colonization of island mountains. Our study explains much of the distributional variation within a complex biological system, and provides a synthesis of two seemingly discordant hypotheses for montane community formation.

Mountains harbor a disproportionately great amount of earth's terrestrial biodiversity[1]. Much of this montane diversity is concentrated in the tropics, where mountains' steep climatic gradients promote high elevational community turnover and endemism[2]. In the face of vast anthropogenic clearance of tropical lowland forests, tropical mountains represent last bastions of intact habitat in many parts of the world[3], but these biotas now face climate change-driven extinction[4,5]. Despite their enormous conservation importance and a long history of intense scientific study[6], we lack answers to basic questions about how montane communities actually form. How does a species arrive on a mountaintop, and what is its fate once it does?

[1]Natural History Museum of Denmark, University of Copenhagen, DK-2100 Copenhagen Ø, Denmark. [2]Centre for Gelatinous Plankton Ecology and Evolution, DTU Aqua, Kemitorvet, Building 202, DK-2800 Kongens Lyngby, Denmark. [3]Center for Evolutionary Hologenomics, Globe Institute, University of Copenhagen, DK-1353 Copenhagen, Denmark. [4]Centre of Excellence for Omics-Driven Computational Biodiscovery (COMBio), Faculty of Applied Sciences, AIMST University, Kedah, Malaysia. [5]Museum für Naturkunde Berlin, Leibniz Institut für Evolutions- und Biodiversitätsforschung, 10115 Berlin, Germany. [6]Animal Ecology, Department of Ecology and Genetics, Evolutionary Biology Centre, Uppsala University, Uppsala, Sweden. [7]Museum Zoologicum Bogoriense, Research Center for Biosystematics and Evolution, National Research and Innovation Agency (BRIN), Cibinong 16911, Indonesia. [8]Department of Bioinformatics and Genetics, Swedish Museum of Natural History, P.O. Box 50007, SE-104 05 Stockholm, Sweden. ✉e-mail: a.reeve@snm.ku.dk

Our understanding of montane biodiversity formation owes much to study within island systems[7–12], and particularly the montane archipelagos of the Indo-Pacific. A striking and long recognized regional pattern is that individual montane bird species are often broadly distributed across many different islands and mountain ranges. The prevalence of these montane supercolonizers inspired the idea that direct colonization between mountain ranges is the key driver of montane community buildup[10,13–15]. The opposite conclusion was reached by E. O. Wilson[7,8], who argued that montane species form via 'taxon cycles' whereby dispersive lowland radiations gradually contract into the highlands to form endemic species restricted to single islands. These ideas are hard to reconcile, and molecular studies have found evidence of both processes[16–20], even within a single mountain community[21].

We investigated whether the variation in observed patterns can be explained by the ancestral source region of montane populations. Passerine birds have entered the many islands scattered between Eurasia and Australo-Papua from one or the other continental source. Lineages from temperate regions may be more successful than tropical lineages at achieving broad and rapid colonization across mountain ranges[21–25]. We hypothesized that Eurasian-origin species, with pre-adaptations to cold and seasonal climates (e.g., migration), colonize directly between island mountains, but that Australo-Papuan-origin species do not, and instead colonize mountains via recruitment from local lowland populations.

Our study finds strong support for these hypotheses. We performed phylogenetic analyses to resolve the evolutionary relationships of the montane passerine avifauna of Wallacea and the Bismarck and Solomon Archipelagos (Fig. 1). We reconstructed the ancestral geographic range, elevational range, and migratory behavior of each species, and analyzed these together with distributional datasets that we compiled. To assess the processes inferred from species-level analyses, we devised and implemented a method to extract and compare homologous gene regions and conducted detailed phylogenomic population studies of montane supercolonizers representing three of the geographically broadest avian montane radiations in the region (Fig. 1). We conducted additional analyses to evaluate whether species' modern distributions are compatible with taxon cycles, how different migratory behaviors correlate with species' success in colonizing island mountains, and whether competitive pressure from close relatives drives montane distributions. Our results provide evidence that the establishment of montane island populations follows fundamentally different processes depending on the long-term region-specific evolution of the parent lineage.

## Results

### Geographic origins of montane island populations

We identified 237 montane island populations (MIPs) of passerine birds distributed across 31 islands in Wallacea and the Bismarck and Solomon Archipelagos, representing 110 species, 47 genera, and 22 families (Table 1; Supplementary Table 1; Supplementary Data 1). Of the 110 species with MIPs, 80 were placed phylogenetically (Table 1; Supplementary Table 1). The 80 species were analyzed within the context of 23 different clades, for which we used five tree files from published studies, one publicly available alignment file, and 17 trees generated specifically for this study (Supplementary Data 2). Clade limits were defined with the goal of determining Eurasian or Australo-Papuan origin (or lack thereof) for species with MIPs. The clades as defined were ~5–15 million years (Ma) old. Each encompassed no more than one family. The 23 clades contain 787 ingroup tips representing 695 species. Ingroup species per clade range from 4 to 185 (mean = 30), and mean species sampling completeness per clade is 90%. The phylogenetic hypotheses generated for this study were highly congruent with the latest published molecular analyses for the relevant groups. Tree files are in Supplementary Data 3. We used these trees to

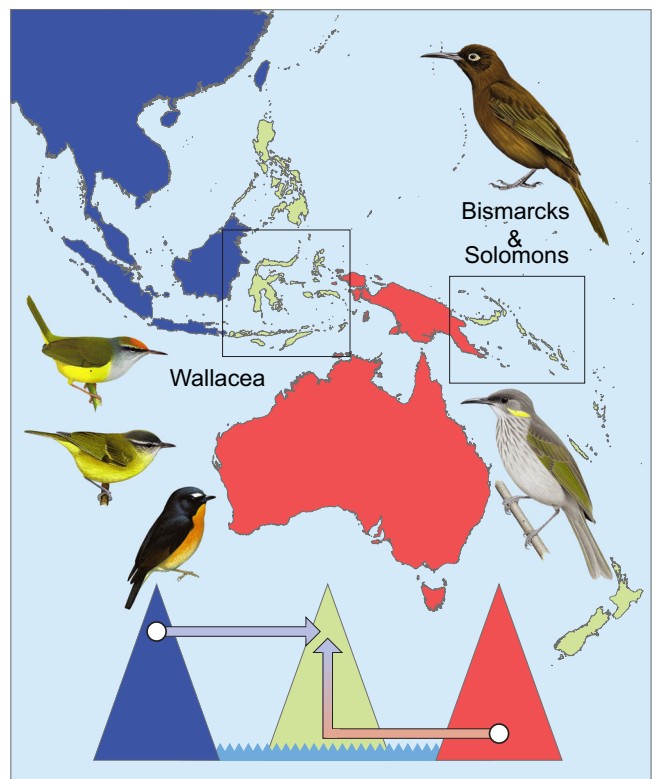

**Fig. 1 | Map of the study region with focal archipelagos highlighted.** The colonization source pools of Eurasia and Australo-Papua are colored blue and red, respectively. The diagram illustrates the respective processes by which Eurasian and Australo-Papuan bird lineages colonize island mountains. At left are three Eurasian-origin montane supercolonizers for which we performed detailed population studies: Mountain Tailorbird *Phyllergates cucullatus*, an Indo-Pacific leaf warbler clade (shown is *Phylloscopus maforensis ceramensis*), and Snowy-browed Flycatcher *Ficedula hyperythra*. At right are two single-island endemic honeyeaters representing Australo-Papuan species. (illustrations by N. Arlott, R. Hathway, D. Quinn, J. Wilczur, and T. Worfolk, from Lynx Edicions).

### Table 1 | Species with montane island population (MIPs) in the focal archipelagos, and summary statistics of those MIPs

|  | All archipelagos | Wallacea | Bismarcks/ Solomons |
|---|---|---|---|
| No. species identified | 110 | 79 | 34 |
| No. species placed in our phylogenies | 80 | 56 | 26 |
| Eurasian species | 31 | 24 | 8 |
| Australo-Papuan species | 25 | 11 | 14 |
| Other species | 24 | 21 | 4 |
| No. MIPs identified | 237 | 176 | 61 |
| No. MIPs represented in our phylogenies | 174 | 131 | 43 |
| Eurasian MIPs | 100 | 79 | 21 |
| Australo-Papuan MIPs | 33 | 17 | 16 |
| Other MIPs | 41 | 35 | 6 |

Note that some species have MIPs both in Wallacea and in the Bismarcks/Solomons.

perform ancestral state reconstructions of geographic range, elevational range, and migratory behavior (Supplementary Table 2; Supplementary Data 4).

We defined the geographic origin of the 80 focal species using ancestral ranges inferred with BioGeoBEARS (v.1.1.2[26,27]). We counted back from terminal tree nodes until reaching "Ancestral Source Nodes"

with >75% probability of being either Eurasian (Palearctic + Indomalaya) or Australo-Papuan (see Methods: Ancestral state reconstructions). 31 Eurasian-origin species represent 100 MIPs from 11 genera and 7 families; 25 Australo-Papuan-origin species represent 33 MIPs from 14 genera and 5 families; and 24 species not clearly tracing back to either focal source area represent 41 MIPs from 14 genera and 11 families (Table 1; Supplementary Table 1; Supplementary Data 1). Locating Ancestral Source Nodes involved counting back 1–10 nodes (mean = 4.2) from tips. Thirty-four unique Ancestral Source Nodes were identified from among the 56 Eurasian-origin and Australo-Papuan-origin species (some species traced back to the same continental ancestor). Mean age of Ancestral Source Nodes for all species is 7.7 Ma.

The Eurasian ancestral source pool of island colonists includes both temperate elements (e.g., Sino-Himalayan mountains) and tropical elements (e.g., South and Southeast Asia). We hypothesized that lineages with temperate origins and preadaptations to cold and seasonal environments are better able to directly colonize island mountains. Eurasian-origin species generally trace back to ancestors with reconstructed distributions spanning both Indomalaya and the Palearctic (Supplementary Fig. 1). Indomalaya is the geographically proximate species source pool feeding the archipelagos. The Indomalayan signature varies, but is >50% for most nodes older than 5 Ma. The root nodes of all Eurasian clades have a 75–100% probability of Indomalayan occurrence, except for *Ficedula* (Muscicapidae; c. 40%). The Palearctic signature either rises markedly (though variably) back through time, or is consistently high (>75%). The root nodes of all Eurasian clades have a 75–100% probability of Palearctic occurrence, except for *Brachypteryx* (Muscicapidae; c. 60%).

## Montane vs. lowland ancestry

Eurasian-origin species with MIPs evolved from predominantly montane continental ancestors (median probability of montane ancestry 1.00), whereas Australo-Papuan-origin species evolved from predominantly lowland ancestors (median probability of montane ancestry 0.08) (Fig. 2). Median probabilities of montane ancestry for

the two groups differ significantly (Mann-Whitney *U*-test: $W = 738$, $p < .001$).

## Establishing multiple MIPs

We tallied the number of MIPs for each species, and tested whether Eurasian-origin and Australo-Papuan-origin species differ in their capacity to establish multiple MIPs. Species dispersing from the respective source regions into the archipelagos have had different possibilities for island colonization and MIP formation. Important varying factors include the number, sizes, heights, and geographic configuration of proximate islands, and past inter-island connectivity via land bridges. To help control for these differences, we additionally made separate MIP counts for Wallacea and Bismarcks/Solomons, and conducted analyses both for all islands, and for the separate island groups. Eurasian-origin species have a higher number of MIPs than Australo-Papuan-origin species (Mann-Whitney *U*-test: $W = 581.5$, $p < .001$) (Fig. 3; Supplementary Tables 3 and 4). This is also the case in separate regional analyses of Wallacea and the Bismarcks/Solomons ($p < .05$ in both cases). A higher proportion of Eurasian-origin species have multiple MIPs (0.65) than do Australo-Papuan-origin species (0.20); $X^2$ (1, $N = 56$) = 9.4, $p < .01$. For individual regions, this relationship is statistically significant in the Bismarcks/Solomons ($p < .05$), but not in Wallacea ($p > .05$).

Species with high numbers of MIPs often have continental populations in Eurasia, an observation which may be useful for understanding the processes by which Eurasian-origin MIPs form (only two Australo-Papuan-origin species have both MIPs and continental populations). To quantify this pattern, we tested whether species with both continental and island populations have more MIPs than species entirely restricted to islands. We found that Eurasian-origin species with continental populations ($n = 13$) have higher numbers of MIPs than species restricted solely to islands ($n = 18$) (Mann-Whitney *U*-test: $W = 182.5$, $p < 0.01$; Supplementary Fig. 2).

## MIPs and lowland island populations (LIPs)

Montane diversity on tropical islands may be the result of taxon cycles[7–9,12]. According to this idea, lowland island radiations gradually adapt to forests of island interiors, shift elevational distributions upwards, experience extinctions in smaller islands, and relictualize as widely separated montane species. Ongoing taxon cycles would presumably leave a signature in the proportion of MIPs to lowland island populations (LIPs). For a given lineage, early stages of taxon cycles should be numerically dominated by LIPs, with relatively few MIPs. As cycles progress, numbers of LIPs and MIPs should become more similar, assuming attainment of montane ranges is not perfectly synchronized across populations. In the late stages of taxon cycles, single-island endemics should form, at which stage the MIP/LIP signature becomes non-informative, as it is identical to a species that has evolved via direct colonization by a montane ancestor.

We examined elevational niche conservatism and looked for evidence of taxon cycles by analyzing the proportions of MIPs to total island populations (MIPs + LIPs) for species of Eurasian-origin and Australo-Papuan-origin, respectively. We predicted a higher proportion for Eurasian-origin species, as a result of their hypothesized direct colonization between island mountains, and a lower proportion for Australo-Papuan species, reflecting ongoing processes of upward range shifts following lowland colonizations. The proportion of MIPs to total island populations (MIPs + LIPs) is indeed higher for Eurasian-origin species (median = 1.00) than for Australo-Papuan-origin species (median = 0.42); median values differ significantly (Mann-Whitney *U*-test: $W = 194$, $p < 0.001$) (Supplementary Fig. 3).

Eurasian-origin and Australo-Papuan-origin species show very different patterns in numbers of MIPs vs. total island populations (Fig. 4). Eurasian-origin species maintain consistently montane distributions as they radiate across archipelagos. This pattern does not fit

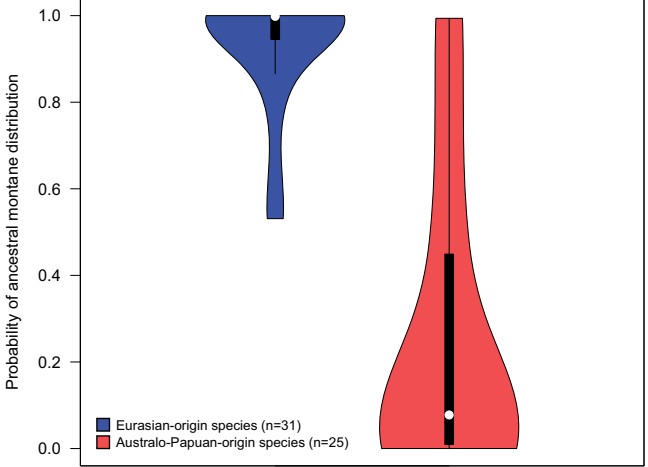

**Fig. 2 | Probability of ancestral montane distribution for Eurasian-origin versus Australo-Papuan-origin species with montane island populations (MIPs).** The violin plots show full data distributions superimposed over standard boxplots. Each box indicates the interquartile range (IQR), with bottom and top edges representing the first (Q1) and third quartiles (Q3), respectively. The median (Q2) is indicated with a white circle. Whiskers extend to minima and maxima not exceeding 1.5 times the IQR below Q1 or above Q3. Values are derived from the probabilities at the Ancestral Source Nodes of all species from the respective groups, as described in Methods: Montane vs. lowland ancestry. Source data for the figure can be found in Supplementary Data 1.

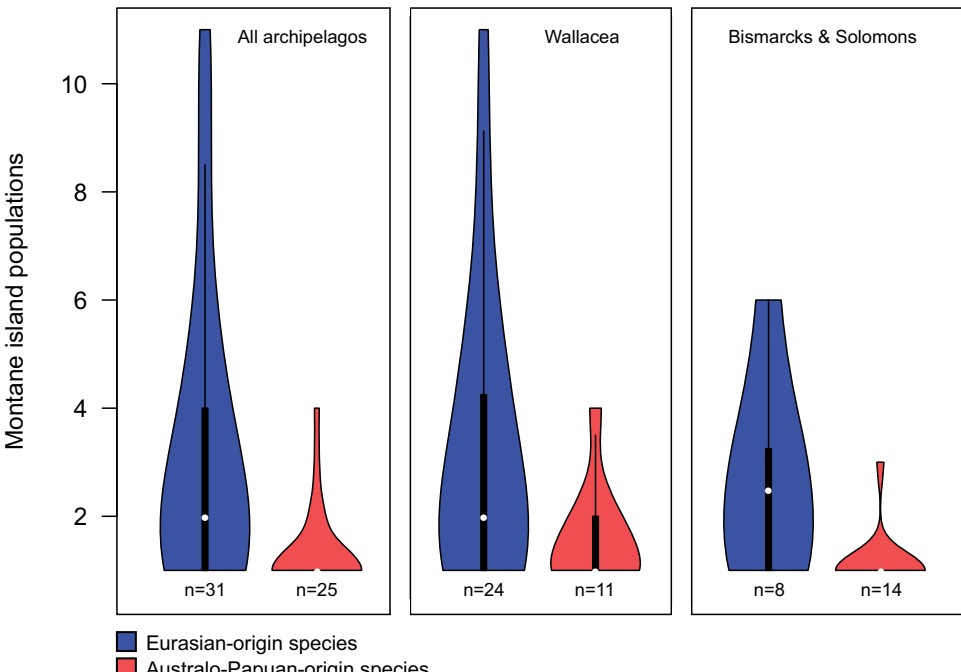

**Fig. 3 | Number of montane island populations (MIPs) per species, by region, for Eurasian-origin versus Australo-Papuan-origin species.** The violin plots show full data distributions superimposed over standard boxplots. Each box indicates the interquartile range (IQR), with bottom and top edges representing the first (Q1) and third quartiles (Q3), respectively. The median (Q2) is indicated with a white circle. Whiskers extend to minima and maxima not exceeding 1.5 times the IQR below Q1 or above Q3. The numbers of species included in each sample are given below the respective plots. Source data for the figure can be found in Supplementary Data 1.

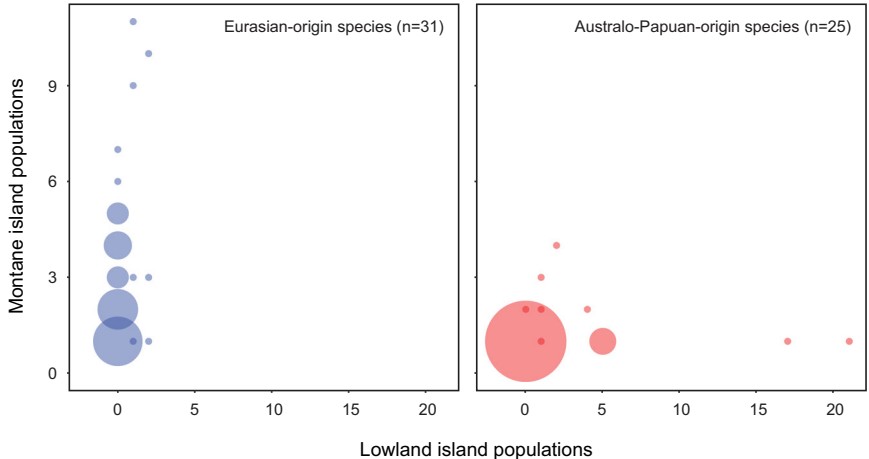

**Fig. 4 | Bubble plots showing number of montane island populations (MIPs) vs. lowland island populations (LIPs) for each individual focal species.** Bubble diameter reflects the number of species sharing specific counts of MIPs vs. LIPs, from one species (smallest bubbles) to 15 species (largest bubble); scale is consistent between the two plots. At left: Eurasian-origin species are consistently montane across their archipelagic ranges. At right: Australo-Papuan-origin species show mixed montane and lowland distributions; the overall pattern could reflect taxon cycles at different stages (see Results: MIPs and lowland island populations [LIPs]). Source data for the figure can be found in Supplementary Data 1.

with conventional taxon cycle theory. There are few examples of species with low proportions of MIPs to total island populations which might represent early- or mid-stage taxon cycles. The species with high proportions of MIPs to total island populations cannot be considered to represent late-stage taxon cycles because they have high numbers of MIPs. In contrast, Australo-Papuan-origin species show an overall pattern that is entirely compatible with taxon cycles. The hypothesized signatures of early stages (many lowland populations), middle stages (a mix of lowland and montane populations), and late stages (one or few montane populations with no lowland populations) are all

represented. Note that although we here treat a taxon cycle as a process leading to a definite end point (extinction of relictual montane species), some conceptions of the taxon cycle also allow for the initiation of new cycles within in-progress cycles. This could complicate prediction of distributional patterns, but presumably any new lowland expansion would result in a species' LIPs outnumbering its MIPs. Australo-Papuan-origin species that are entirely montane within the focal archipelagos are older than those having both MIPs and LIPs, in accordance with predictions of the taxon cycle. We tested this by defining species age as the estimated time to divergence from a sister

species or clade in our phylogenetic trees. The average node age of species where the proportion of MIPs to total island populations = 1 ($n = 16$; median = 5.46 Ma) is significantly higher than average node age of species where that proportion <1 ($n = 9$; median = 1.30 Ma) (Mann-Whitney $U$-test: $W = 22$, $p < 0.01$).

## Migration

Lineages from regions with cold and seasonal climates appear better able to make broad and rapid colonization across mountain ranges than tropical lineages[21–25]. Any specific traits of temperate lineages that promote colonization across mountain ranges presumably vary between organism groups. In birds, an obvious candidate trait linked with mobility is migratory capacity. We tested whether the Eurasian ancestors of species with MIPs are in fact more migratory than Australo-Papuan ancestors. Although migration is widespread in Eurasia, lineages in South and Southeast Asia (which directly feed the focal archipelagos) show very little migratory behavior.

More than half of the ancestors of the 31 Eurasian-origin species were inferred to have short-distance migrant populations, and very few had long-distance migrant populations (Supplementary Fig. 4; Table 2). Most had at least some sedentary populations across their ranges. The ancestors of the 25 Australo-Papuan-origin species were almost entirely sedentary, with very few short-distance migrant populations and no long-distance migrant populations (Table 2). Comparison of median ancestry probabilities (0.64 for Eurasian-origin ancestors, 0.00 for Australo-Papuan ancestors) confirmed that the former had a higher probability of making migratory movements (Mann-Whitney $U$-test: $W = 744$, $p < .001$).

Species' ability to form multiple MIPs may be linked to their capacity for short-distance migration, specifically; some of the most effective colonizers of island mountains have short-distance migrant populations within their global ranges. Our tests confirmed that species with short-distance migrant populations have higher numbers of MIPs than species that are sedentary throughout their range (Fig. 5). Short-distance migrant populations occur only outside the focal archipelagos, and no species with MIPs have long-distance migrant populations When considering all species with MIPs, species with short-distance migration ($n = 12$) have a median of 5.50 MIPs, and entirely sedentary species ($n = 98$) have a median of 1.00 MIPs. Median values for the two groups differ significantly (Mann-Whitney $U$-test: $W = 142.5$, $p < .001$). Among Eurasian-origin species, species with short-distance migration ($n = 8$) have a median of 4.50 MIPs, and entirely sedentary species ($n = 23$) have a median of 2.00 MIPs. Again, median values for the two groups are significantly different (Mann-Whitney $U$-test: $W = 147$, $p < .05$).

## Competitive pressure in the lowlands

Competitive pressure from closely related lowland species is often implicated in the formation of new montane populations from lowland source pools[28]. This mechanism may not be important for species colonizing directly between mountains and retaining their elevational niche. In line with these hypotheses, Australo-Papuan-origin MIPs often share islands with lowland relatives while Eurasian-origin MIPs do not. To quantify this pattern, we determined whether individual MIPs share islands with breeding species from the same genus and family (respectively) occurring in the lowlands. We confirmed that a higher proportion of Australo-Papuan-origin MIPs face potential competition from lowland congeners (0.33) than do Eurasian-origin MIPs (0.06) ($X^2$ (1, $N = 133$) = 14.3, $p < .001$). Similarly, a higher proportion of Australo-Papuan-origin MIPs face potential competition from lowland species from the same family (0.73) than do Eurasian-origin MIPs (0.36) ($X^2$ (1, $N = 133$) = 12.1, $p < .001$). Nevertheless, competition from lowland

## Table 2 | Ancestral migratory behavior of Eurasian-origin vs. Australo-Papuan-origin species with MIPs

|  | Sedentary | Short-distance migrant | Long-distance migrant | Any movement |
|---|---|---|---|---|
| Eurasian species | 0.86 | 0.64 | 0.03 | 0.64 |
| Australo-Papuan species | 1.00 | 0.00 | 0.00 | 0.00 |

Numbers are the median probabilities of migratory behavior classes at Ancestral Source Nodes for all species. Migratory behavior classes are not mutually exclusive for species or ancestral nodes, as different populations within a single species can show different migratory behavior.

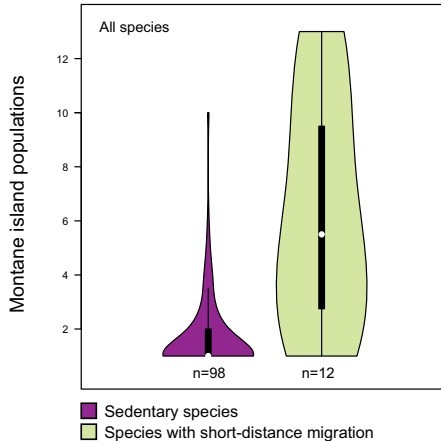
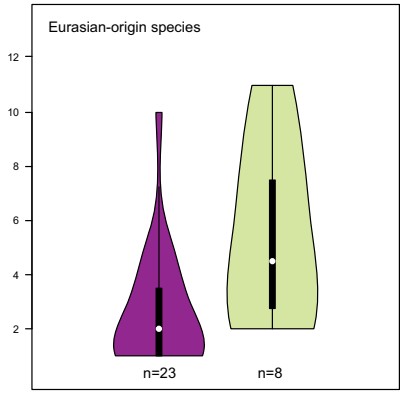
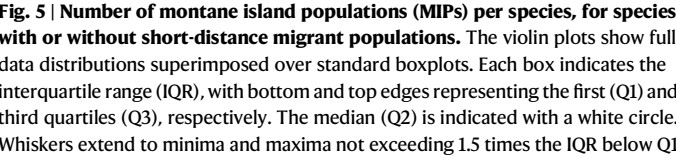

**Fig. 5 | Number of montane island populations (MIPs) per species, for species with or without short-distance migrant populations.** The violin plots show full data distributions superimposed over standard boxplots. Each box indicates the interquartile range (IQR), with bottom and top edges representing the first (Q1) and third quartiles (Q3), respectively. The median (Q2) is indicated with a white circle. Whiskers extend to minima and maxima not exceeding 1.5 times the IQR below Q1 or above Q3. Short-distance migrant populations occur only outside the focal archipelagos, and no species with MIPs have long-distance migrant populations. At left: all 110 species with MIPs in the focal region, including species we did not genetically sample. At right: Eurasian-origin species only. The numbers of species included in each sample are given below the respective plots. Source data for the figure can be found in Supplementary Data 1.

relatives does not appear to be the sole factor driving MIP formation for Australo-Papuan-origin species, because a number of Australo-Papuan MIPs lack close relatives in the lowlands. The higher number of lowland relatives found for Australo-Papuan MIPs does not correlate with overall higher levels of diffuse competition experienced by those populations; there is no significant difference in the size of the lowland passerine island assemblages faced by Australo-Papuan- versus Eurasian-origin MIPs (Mann-Whitney $U$-test: $W = 1361$, $p = 0.13$). We stress that the results presented in this section indicate only the potential for competition from lowland relatives. Direct evidence requires field experiments[29,30].

## Phylogenetic null models

We used phylogenetic null models to assess how well the empirical patterns of the above statistical tests could be replicated based on data simulated under a Brownian motion model of evolution (Supplementary Table 5). These tests include comparisons of Eurasian- versus Australo-Papuan-origin species: montane versus lowland ancestry, number of MIPs (both total and for individual regions), proportion of MIPs to total island populations, and migration ancestry. We also tested, for Eurasian-origin species, number of MIPs for short-distance migrant versus sedentary species, and number of MIPs for species with continental populations versus those restricted entirely to islands. The phylogenetic null models were performed to determine how well our results are explained by phylogenetic history alone, in the absence of direct quantification of geographic or ecological characters among lineages. For most of the aforementioned analyses, our empirical F statistics are significantly greater than those derived from the simulated null datasets. The F statistic represents the ratio of the variance between and within groups; the value becomes higher as the variance between groups increases relative to the variance within them. When the empirical value of F is greater than the distribution of values simulated under Brownian motion, we conclude that the patterns in these particular variables are greater than would be predicted based purely on phylogeny. However, the particular pattern that Eurasian-origin species have more MIPs than Australo-Papuan-origin species can be reproduced upon simulating the data using Brownian motion alone. This is true with regard to all islands ($p = .07$), for Wallacea alone ($p = .16$), and for Bismarcks/Solomons alone ($p = .08$).

One potential source of this strong phylogenetic signal is the tight phylogenetic clustering of certain species with MIPs, most obviously the Indo-Pacific *Phylloscopus* leaf warbler clade that has spawned 11 MIPs in just four million years (see Results: Population studies of montane supercolonizers). However, this does not seem to be the source of the signal, because the average number of MIPs within these species clusters (e.g., a mean of 2.1 MIPs per species across all islands in the Indo-Pacific *Phylloscopus* leaf warbler clade) is similar to the average across all species within their respective geographic groups (mean = 3.2 MIPs per Eurasian-origin species across all islands). Rather, the source of the phylogenetic signal appears to derive from the deep structure of the relationships of all species in the analysis (Fig. 6). It is clear that our comparison of Eurasian and Australo-Papuan avifaunas is nearly equivalent to a comparison of three major passerine clades: Passerides (Eurasia), Corvides (Australo-Papua), and Meliphagides (Australo-Papua). A strong phylogenetic signal is therefore inherent to our study.

## Population studies of montane supercolonizers

Among the birds inhabiting the focal archipelagos, certain species have an extraordinarily high capacity to disperse between and colonize the mountains of different islands. Reconstructing the evolutionary histories of these 'montane supercolonizers' can shed light on how lineages entered archipelagic mountains, and provide insight into the nature of mountain-to-mountain dispersal. We made detailed population studies of three species or clades representing montane

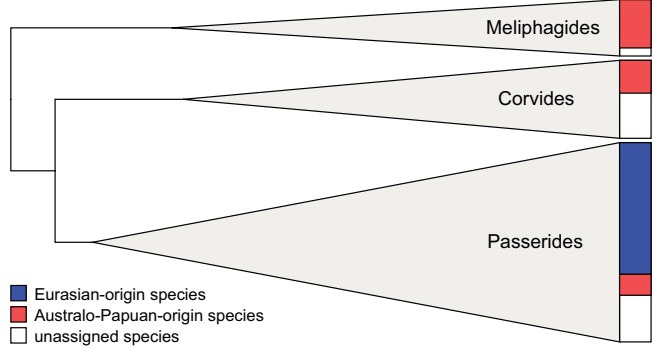

**Fig. 6 | Evolutonary relationships of the 80 species with montane island populations (MIPs) placed within a phylogenetic context in this study.** Species are distributed across three passerine clades, represented by gray triangles. The triangles are scaled proportionally to the number of species with MIPs in each clade. The colored bars to the right show the proportion of Eurasian- versus Australo-Papuan-origin species in each clade. White rectangles represent 24 species which did not trace back clearly to Eurasia or Australo-Papua. Source data for the figure can be found in Supplementary Data 1 and 3.

supercolonizers: Indo-Pacific *Phylloscopus* leaf warblers (Phylloscopidae), Snowy-browed Flycatcher *Ficedula hyperythra* (Muscicapidae), and Mountain Tailorbird *Phyllergates cucullatus* (Cettiidae) (Fig. 1). These are small (6–12 g) insectivores with ranges spanning continental Eurasia and Indo-Pacific archipelagos, showing very subtle morphological differentiation between populations. They were chosen because they are among the most successful colonizers of island mountains in the focal region, and because it was possible to confidently infer their continental source pools – available evidence[31–33] suggests Eurasian ancestral origins for all three. Population studies of Australo-Papuan-origin passerines with broad island distributions but few MIPs can be found in Andersen et al.[34] and Jønsson et al.[18].

The population studies were designed to answer several key questions. First, we wanted to confirm that the species/clades in question are actually monophyletic, representing single radiations. Modern molecular work regularly reveals Indo-Pacific island "species" to be aggregations of similarly-plumaged but unrelated populations (see e.g. Andersen et al.[34] and Moyle et al.[33]). The primary goal, however, was to investigate whether the processes inferred at species level for mountain-to-mountain colonizers could be seen in individual island radiations. Are continental occurrence, montane distribution, and migratoriness ancestral conditions that existed prior to species' expansions into archipelagos, or vice versa? Are montane island radiations from continental Asia the product of west-to-east 'stepping-stone' dispersal, or do modern island populations represent previously migratory populations that have become resident? And how did radiations proceed through time?

Twelve trees representing four phylogenetic analyses for each of the three montane supercolonizers are presented in Supplementary Figs. 5–16. Discussion of the results centers around the dated supermatrix analyses, which have near-complete taxon coverage, and the genomic maximum likelihood (ML) trees based on concatenated data, which are highly resolved. The genomic species trees are entirely congruent with the genomic ML trees, but support values are slightly lower. Mitogenome trees are variously well resolved, and in some cases recover highly supported relationships conflicting with the other genomic analyses, which may reflect introgression between populations[35].

The groups analyzed are mostly monophyletic and represent individual island radiations, although the single-island endemic Damar Flycatcher *Ficedula henrici* is embedded within Snowy-browed Flycatcher. Overall, the population studies reinforce the results obtained using the species-level ancestral state reconstructions in showing that

the three montane supercolonizers evolved from continental Asian ancestors that were montane with a tendency for short-distance migration. The analyses strongly support the idea that all three radiations proceeded eastwards out of continental Indomalaya following a stepping stone pattern of sequential island colonization. All radiations included separate expansions (1) across the Lesser Sundas, and (2) via Sulawesi into the Moluccas (and in the case of the Indo-Pacific *Phylloscopus* leaf warblers, further into Melanesia). Radiations varied in tempo and duration, but took place mostly or entirely within the Pleistocene. The small number of lowland populations of montane supercolonizers were found to represent a few individual shifts away from an otherwise montane distribution. Two populations of altitudinal migrants belong to early branching clades within their respective groups, further suggesting an ancestral propensity for short-distance migration among montane supercolonizers. For more information see Supplementary Note 1: Detailed results of population studies of montane supercolonizers.

## Discussion

Our study provides a synthesis that reconciles contrasting hypotheses about the formation of Indo-Pacific montane diversity. We find that montane island populations' (MIPs) modes of colonization, and their potential for further montane colonizations, are governed by the long-term region-specific evolution of their parent lineages. Eurasian-origin MIPs derive from montane continental ancestors, while Australo-Papuan-origin MIPs evolved from lowland continental ancestors. Montane distribution does not inhibit further montane colonization by Eurasian-origin species, but it does for Australo-Papuan-origin species. Here we seek to characterize and explain these dichotomous processes.

This study compares Eurasian and Australo-Papuan avifaunas, but it is also a comparison between avifaunas of temperate (Eurasian) versus tropical (Australo-Papuan) climates. This may not be immediately intuitive, because tropical Southeast Asia serves as a gateway for colonization of the focal islands. However, our ancestral geographic and elevational range reconstructions indicate that virtually none of the montane diversity in the focal islands derives from lowland Indomalaya (Fig. 2; Supplementary Data 4). The ancestors of Eurasian-origin MIPs occurred in cool montane climates (Fig. 2) with ranges spanning both the Palearctic and Indomalayan realms (Supplementary Fig. 1). Lowland Indomalayan lineages, on the other hand, may actually generate montane populations within Indomalaya itself in a similar way to lowland Australo-Papuan lineages; studies from the Greater Sundas provide examples of range-restricted montane endemics arising via recruitment from regional lowland clades[21], sometimes driven by displacement by lowland congeners[36,37]. However, lowland Indomalayan lineages differ strikingly from lowland Australo-Papuan lineages in their near complete failure to generate MIPs in the focal archipelagos.

Montane island populations experience different fates depending on their ancestral origins. For Eurasian-origin species, mountains often represent stepping stones to additional montane colonizations. Our analyses indicate that they retain an ancestral montane niche as they disperse, often extensively, across archipelagos (Figs. 2–4; Supplementary Figs. 5–16). Conversely, Australo-Papuan-origin species appear to have arisen via past island colonizations by lowland ancestors (Fig. 2), and little in their modern distribution patterns indicates an ability to colonize directly between the mountains of different islands (Figs. 3 and 4; see also Results: MIPs and LIPs). For these species, mountains represent dead ends for further dispersal. Both Eurasia and Australo-Papua have substantial montane avifaunas distributed across broad elevational gradients. Australo-Papuan montane species can colonize the disjunct mountain ranges of New Guinea[15,38], but they are extremely poor at crossing deep water barriers. The different colonization processes of Eurasian-origin versus Australo-Papuan-origin birds appears to be rooted in the temperate/tropical dichotomy in our study system, aligning with previous observations that lineages from regions with cold and seasonal climates are adept at colonizing disjunct mountain ranges[21–25]. Eurasian montane species, with high diversity in the Sino-Himalaya, face much more extreme annual climate variation than their Australo-Papuan counterparts concentrated near the equator in New Guinea. This contrast has existed for millions of years, and has promoted mobility in the form of elevational migration among Eurasian montane species (Table 2; Supplementary Fig. 4). We show that a capacity for short-distance migration can help predict whether a species or clade experiences a mountaintop as a stepping stone or a dead end (Fig. 5).

It may be conserved morphological or cognitive adaptations for short-distance migration, or both, that aid establishment of multiple MIPs. Ancestral short-distance migration specifically − as opposed to long-distance migration—may be an important promoter of mountain-to-mountain colonization. Our reconstructions show virtually no trace of long-distance migration among the ancestral source nodes of species with MIPs (Supplementary Fig. 2), and long-distance migration has been shown to actually constrain colonization of new regions[39]. Nomadism driven by granivory, nectarivory, or frugivory may provide a similar catalyst. The few non-passerines that have colonized mountains on multiple islands in the focal archipelagos are parrots (*Charmosyna*, *Micropsitta*) and pigeons (*Gymnophaps*) that make nomadic wanderings in search of fruit or flowers[40,41]. Short-distance migration may be linked with an ability to rapidly and flexibly track shifting resources, facilitating colonization of island mountains; however, relationships between niche breadth, dispersal capacity, and range size in short-distance migrants are not well understood[42,43].

Our phylogenetic null models (Supplementary Table 5) underscore that our comparison of Eurasia vs. Australo-Papuan birds is, to a high degree, a comparison between the large passerine clades Passerides (Eurasia), and Corvides and Meliphagides (Australo-Papua) (Fig. 6). All three clades initially evolved in Australia, but Passerides quickly expanded into Eurasia c. 30 Ma and then radiated across the globe[44]. This raises the intriguing possibility that Eurasian species' high propensity for mountain colonization is related to ancestral traits of Passerides that evolved prior to, or even facilitated the clade's initial expansion into Eurasia. The global expansion of Passerides has been attributed to a possible early evolution of increased thermal flexibility and short-duration social pairing for breeding (reviewed in Christidis et al.[45]). However, as we demonstrate, the ability to colonize directly between mountains is disproportionately manifested among species that come from a specific temperate evolutionary background, and that have behavioral traits reflective of that ancestry (i.e., short-distance migration). This speaks strongly for region-specific selective pressures being the causative mechanism.

Why species acquire and retain montane distributions remains an open question[46], but our study system provides some insights. Competition from closely related species may drive upward range shifts in Australo-Papuan-origin MIPs, which often share islands with lowland species from the same genus and/or family. This is not the case for Eurasian-origin MIPs, which mostly do not have close relatives in the lowlands (see Results: Competitive pressure in the lowlands). Clues about the mechanisms driving montane distributions for Eurasian-origin MIPs can be found by looking at their conspecific populations in continental Asia. Species with altitudinal migrant populations in the Sino-Himalayan region are among the most successful mountain-to-mountain colonizers in the focal archipelagos. These Sino-Himalayan populations spend much of the year in the lowlands, indicating that montane distribution is not a physiological imperative, nor is it driven by susceptibility to lowland pathogens[47]. Rather, mid-elevation peaks in arthropod abundance may drive songbirds' montane distributions in that region[48]. Diamond[49] argues that Island Thrush *Turdus poliocephalus*, a Eurasian-origin montane supercolonizer, has an elevational

range that is governed by levels of diffuse competition on different islands; i.e., it is a weak competitor that is restricted to mountains where overall bird species diversity is high, but occurs into the lowlands on species-poor islands. This explanation cannot be applied generally to the entire assemblage of Eurasian-origin species with MIPs, nearly half of which have tropical and subtropical continental populations from species-rich mid-montane communities. Further, if diffuse competition drives montane distributions, it is unclear why montane species have not colonized the myriad small, low, species-poor islands in the focal region (Supplementary Data 1). Avoidance of high nest predation pressure in the lowlands (e.g., from *Boiga* snakes) may be a more parsimonious explanation for these patterns, and is a mechanism that is increasingly thought to drive montane species distributions[46,50,51].

There is no single agreed-upon definition of taxon cycles[8,12], and arguments for their existence necessarily hinge upon inference of range shifts, niche shifts, and extinctions which mostly leave no direct evidence. However, the patterns we recover among Australo-Papuan-origin species fit with its basic idea that broad lowland radiations colonize archipelagos, gradually move into the mountains of larger islands while experiencing extinctions on smaller islands, and leave behind relictual species restricted to the mountains of single islands. We recover the hypothesized lowland-to-highland distributional shift (Fig. 2), and find that this coincides with a diminished capacity for further colonization (Fig. 3). Australo-Papuan-origin species never have many montane populations in the focal archipelagos, but can have few or many lowland populations (Fig. 4; Supplementary Fig. 3), and this may reflect taxon cycles in varying stages of advancement. Eurasian-origin MIPs clearly did not arise by this process; they retain an ancestral montane distribution as well as the ability to colonize additional island mountains (Fig. 4; Supplementary Fig. 3). Eurasian-origin archipelagic radiations do appear to slow down over time, based on the observation that species with continental occurrence have more MIPs than species restricted to archipelagos (Supplementary Fig. 2). Whether this slowing predictably leads to lineage relictualization, either in mountains or in lowlands, is unclear.

The taxon cycle idea predicts consistent upward range shifts over time, and this has the potential to confound our elevational range reconstructions of ancestral source nodes. If a clade consists mostly of montane late-stage taxon cycle relicts, then its deeper nodes will be incorrectly reconstructed as montane. This issue mostly does not manifest itself in the reconstructions of Australo-Papuan groups (Supplementary Data 4) because modern lowland distribution is so pervasive across the relevant clades. It is more pertinent for the reconstructions of Eurasian groups (Supplementary Data 4), which indicate montane distributions for virtually all ancestral source nodes (Fig. 2). Nevertheless, it does not seem plausible that this issue has biased the results because that would require that nearly all MIPs and the species in their constituent clades represent late-stage taxon cycles, when in fact many clades are recently evolved and speciose, and contain species with broad montane distributions. Further, our population studies of montane supercolonizers (Supplementary Figs. 5–16) reveal formation of montane populations until very recent times, so taxon cycles would have to be extremely fast as well as highly synchronized. The parsimonious explanation for this pattern is that species retained the montane distribution of their Eurasian ancestors.

Montane island populations face a double barrier to dispersal, with both lowland areas and expanses of water presenting formidable obstacles[13]. However, the overall colder climate and repeated glacial periods in the Pleistocene would have weakened these barriers through the downslope expansion of montane forest[52], and the formation of land bridges between some islands due to lower sea levels. Low intervening islands likely hosted cool forests suitable for montane species, further facilitating their dispersal through archipelagos[53,54]. While cooler climates cannot have completely eliminated lowland competition, predation, and pathogen pressure, the Pleistocene almost certainly brought improved opportunities for colonization between island mountains. Eurasian-origin and Australo-Papuan-origin species reacted very differently to these opportunities, however. Eurasian-origin montane species made numerous colonizations of archipelagos, and established many MIPs during this time. This can be inferred if we assume that most intra-species population divergences are younger than 2.6 Ma. Our population studies of montane supercolonizers (Supplementary Figs. 5–16) support this, indicating that the vast majority of colonizations across deep and shallow water barriers occurred during the Pleistocene. Australo-Papuan-origin species, by contrast, had almost no perceptible reaction to Pleistocene cooling. Only two Australo-Papuan-origin species have continental populations, indicating that there was minimal colonization of archipelagos by montane species in this grouping during this time. There are few species with multiple MIPs, and among these, MIP numbers are low (Fig. 3; Supplementary Tables 3, 4). Therefore, while Pleistocene cooling potentially facilitated inter-island colonization by these species' lowland ancestors, it does not seem to have promoted colonization by montane populations. In addition to the lower dispersal capacity that we infer for Australo-Papuan-origin MIPs, the potentially higher levels of competition they face from lowland relatives may have contributed to preserve the isolation of montane populations during the Pleistocene.

The complex patterns of earth's montane biodiversity have inspired conflicting hypotheses about how it evolved. We reconcile these hypotheses by demonstrating that avifaunas from different continents follow fundamentally different processes to generate montane populations. This dichotomy appears to reflect an important divide between temperate and tropical lineages. As our knowledge of the phylogenetic relationships of species rapidly improves, it will be increasingly feasible to investigate how the region-specific evolution of lineages has shaped contemporary global biodiversity patterns.

## Methods

### Defining montane island populations

In this study we focus on the passerine birds of Wallacea and the Bismarck and Solomon Archipelagos. We follow the conventional modern geographic delimitation of Wallacea as defined by Darlington[55] (see Ali & Heaney[56]). We follow Mayr's and Diamond's[11] delimitations of the Bismarcks and Solomons. Taxonomic classification follows IOC v 9.2[57] unless otherwise noted (see Supplementary Data 5).

We are interested in species with populations that are restricted entirely to the mountains. A centrally important unit in this study is the montane island population ("MIP"), defined as an individual island population that occurs in the mountains but not at sea level (i.e., has not been recorded below 100 masl). This is the definition used by Mayr and Diamond[11] to classify montane populations in Northern Melanesia. Our focus is on populations that cannot persist at sea level, and therefore this definition is not intended to distinguish populations that reach the highlands from those restricted to the lowlands (although it does broadly have that effect). For example, a population occurring at 0–2,000 masl is considered "lowland," while a population occurring at 500–1,500 masl is considered "montane."

We identified MIPs using distributional information from Coates & Bishop[58] for Wallacea, and from Dutson[59] for the Bismarcks and Solomons. This dataset was further refined after a careful review of subsequently published primary literature on the respective regions (Supplementary Data 1). Certain small islands within the focal region probably host one or more MIPs, but were excluded because available distributional data were too limited to accurately identify them. Certain MIPs known from very few records presumably do reach the lowlands, given the species' lowland distributions throughout the rest of their ranges; these were removed from consideration. MIPs of non-

breeding migrants were not considered, and there are no very clear examples of such populations.

## Phylogenetic trees for ancestral state reconstructions

We placed species with MIPs in species-level phylogenetic trees to allow reconstruction of ancestral geographic range, elevational range, and migratory behavior. We obtained tree files from published analyses for some groups, but otherwise produced new trees using both sequence data from GenBank and data newly generated for this study (Supplementary Data 5 and 6). We did not run phylogenetic analyses for clades in which taxonomic coverage of sequence data was very incomplete, or for clades where the latest molecular studies fail to resolve species relationships to a useful degree.

Phylogenetic analyses were performed using 1–6 genes per clade (see Supplementary Data 5). Both nuclear and mitochondrial genes were used for three clades of particular interest (see Methods: Taxon sampling of montane supercolonizers), and for an additional clade for which we obtained a publicly available alignment file. Analyses of the remaining clades were built upon a subset of 1–2 mitochondrial genes, which are phylogenetically highly informative at the recent timescales with which we are concerned. Recent studies have demonstrated varying levels of discordance between phylogenetic analyses driven largely by mitochondrial genes versus those using dense sampling of nuclear markers (e.g., Andersen et al.[60]). Is therefore necessary to recognize a degree of topological uncertainty in the analyses described in this section, even in highly-supported relationships. However, this should not affect tree topologies so as to bias the ancestral state reconstructions and downstream analyses in any particular direction.

Individual gene alignments were built using MAFFT[61], as implemented in SeaView (v.4.6.2[62]). We analyzed individual gene partitions in BEAST (v.1.8.4[63]) as a preliminary quality check of the sequence data. We analyzed the concatenated datasets, partitioned by genes in BEAST, using the GTR nucleotide substitution model for mitochondrial genes (unlinked), and the HKY nucleotide substitution model for nuclear genes (unlinked). We used a relaxed uncorrelated lognormal distribution for the molecular clock model (all genes unlinked), and assumed a Birth-death speciation process as a tree prior. For each clade, the Markov chain Monte Carlo (MCMC) algorithm was run three times for 100 million iterations, with trees sampled every 10,000th generation. Convergence of individual runs was assessed using Tracer (v.1.6[64]), ensuring all ESS > 200, and graphically estimating an appropriate burn-in (10 million generations in most cases). In cases where individual runs failed to converge, nucleotide substitution and molecular clock models were replaced with simpler models. TreeAnnotator (v.1.8.2[65]) was used to summarize a single maximum clade credibility (MCC) tree using mean node heights. To obtain absolute dates, we followed Lerner et al.[66] and applied a rate of 0.007 substitutions per site per lineage (1.4%) per Ma to cytochrome *b* (*cyt-b*) data; 0.008 substitutions per site per lineage (1.6%) per Ma to cytochrome *c* oxidase I (*COI*) data; 0.0145 substitutions per site per lineage (2.9%) per Ma to NADH dehydrogenase 2 (*ND2*) data; and 0.012 substitutions per site per lineage (2.4%) per Ma to NADH dehydrogenase 3 (*ND3*) data.

We compared our trees individually against the most current and comprehensive published trees available for the respective groups, and evaluated congruence between well-supported nodes (posterior probability >0.98) in those trees with our own results. Details on individual analyses are in Supplementary Data 5, including taxonomic coverage per clade, gene sets, departures from default analysis settings, and publications referenced for tree topologies.

## Ancestral state reconstructions

We used the R package BioGeoBEARS[26,27] to infer ancestral geographic ranges, elevational ranges, and migratory behaviors across the generated trees (see Methods: Phylogenetic trees for ancestral state reconstructions). We compared Dispersal-Extinction Cladogenesis (DEC) models[67] with and without an additional free parameter (+j) that allows for founder-effect speciation. These models are typically used for the reconstruction of ancestral geographic ranges, but are also useful for reconstructing ancestral elevational ranges and migratory behavior, as the evolution of these traits parallels the modeled processes of geographic range evolution. We assessed model fit using the Akaike Information Criterion (AIC). We pruned a small number of redundant tips so that a single tip represents each species. Trait scoring drew primarily upon Handbook of the Birds of the World Alive[68], supplemented by regional field guides, primary literature, and in a few cases, critically evaluated eBird[69] records. This information was applied within the framework of IOC v 9.2[57] species limits (except for noted exceptions − see Supplementary Data 5).

Species' geographic ranges were defined as their breeding distributions within nine biogeographic regions (Supplementary Data 4): Palearctic, Indomalaya, Philippines, Wallacea, Australo-Papua, Bismarcks + Solomons, Pacific, Afrotropics, and Americas. Our definition of the ambiguous boundary between the Palearctic and Indomalayan regions follows Udvardy[70], and we set the precise boundary along the Himalayas at the freezing line[71]. We did not score species as inhabiting regions where they occur extremely marginally relative to their overall range. The certainty in nodal states for the reconstruction of *Turdus* (Turdidae) was confounded by the inclusion of a single species (*Turdus poliocephalus*) that occurs across six Indo-Pacific island regions in an otherwise strictly Eurasian clade. To address this, we lumped Philippines, Wallacea, Bismarcks + Solomons, and Pacific into a single region for that particular analysis.

Elevational ranges were defined as "Lowland," "Montane," or "Lowland + Montane" according to species' breeding distributions. Scoring followed essentially the same criteria used to define MIPs (see Methods: Defining montane island populations), though research of elevational limits for each species was by necessity less rigorous than for MIPs in the focal archipelagos. Additionally, species with geographically disjunct lowland and montane populations − whether on different islands or the same landmass − were scored "Lowland + Montane."

Migratory behavior was categorized as "Sedentary," "Short-distance Migration," and "Long-distance Migration." "Short-distance Migration" includes a spectrum of movement from nomadism and altitudinal migration to annual migratory movements up to 2,000 km. "Long-distance Migration" is defined as regular migratory movements over 2,000 km. Species were scored for all categories of movement shown by their constituent populations.

## Geographic origin of species

We attempted to identify respective Eurasian (Palearctic + Indomalaya) or Australo-Papuan ancestral origin for the 80 species with MIPs that were included in geographic reconstructions. We defined individual species' ancestral origins by counting back from terminal tree nodes until reaching Ancestral Source Nodes with > 75% probability of being either Eurasian or Australo-Papuan. Species for which the 75% threshold was not crossed in the reconstructions do not have clear ancestral origins in either source region, and these were not considered further.

We wanted to test the hypothesis that features of the area of ancestral origin influence the pattern and process of MIP formation in archipelagos. We are therefore concerned with ancestral ranges that are largely restricted to (not merely inclusive of) Eurasia or Australo-Papua. This is reflected in our treatment of the reconstructed ancestral range probabilities for this exercise, specifically that we divided composite region scores into their individual constituent parts. For example, if a reconstructed node had a 100% probability of "Australo-Papua + Wallacea", we treated that node as being 50% Australo-Papuan and 50% Wallacean.

To estimate the respective temperate vs. tropical evolutionary backgrounds of Eurasian-origin MIPs, we plotted the posterior probabilities of a Palearctic or Indomalayan ancestral area back through time. For this exercise we did not divide composite region scores as described above.

## Montane vs. lowland ancestry

To infer whether Eurasian-origin and Australo-Papuan-origin species evolved from lowland or montane continental ancestors, and whether there was a difference between the two groups, we used the reconstructed probabilities of montane distribution for the Ancestral Source Nodes of each species. In some cases, models reconstructed simultaneous lowland and montane distributions for the Ancestral Source Node, typically with low probability. Here we split the combined "Montane/Lowland" probabilities evenly, so that 50% of the probability was attributed to the overall "Montane" score, and 50% to the overall "Lowland" score. Mann-Whitney U-tests were used to compare probabilities of lowland vs. montane ancestry between the Eurasian and Australo-Papuan groups.

## Establishing multiple MIPs

We tested for differences in number of MIPs between Eurasian-origin vs. Australo-Papuan-origin species using Mann-Whitney U-tests. We additionally tested for differences in the proportion of Eurasian-origin vs. Australo-Papuan species having >1 MIP using chi-squared tests. We further tested whether species with both continental and island populations have more MIPs than species entirely restricted to islands. Occurrence of Eurasian-origin species in Australo-Papua was not considered to constitute "continental occurrence." Comparison of medians was made using a Mann-Whitney U-test.

## MIPs and LIPs

LIPs were defined as populations with elevational ranges extending below 100 masl within the focal regions. These were identified for all Eurasian-origin and Australo-Papuan-origin species with MIPs, following a process similar to that used for MIP identification (see Methods: Defining montane island populations). Certain island populations were removed from the analysis if no elevational distribution data were available, and the island in question has highlands extensive enough to plausibly support a montane population. We did not include species with only one MIP and no LIPs in the analysis for the reason stated above. Comparison of proportions was performed for Eurasian-origin vs. Australo-Papuan-origin species using a Mann-Whitney U-test.

## Migration

To determine whether Eurasian-origin and Australo-Papuan-origin species evolved from migratory or sedentary continental ancestors, and whether there was a difference between the two groups, we used the reconstructed probabilities of migratory behavior for the Ancestral Source Nodes of each species. We calculated the total probability that the ancestral species showed each category of migratory behavior. For example, the probability of "Sedentary" was calculated by the summing the probabilities of the states Sedentary, Sedentary+Short, Sedentary+Long, and Sedentary+Short+Long. Comparison of median probabilities for Eurasian-origin versus Australo-Papuan-origin species was made using a Mann-Whitney U-test.

We tested whether species with short-distance migrant populations form more MIPs than species that are sedentary throughout their range. These short-distance migrant populations exist exclusively outside of the focal archipelagos. We first analyzed all species with MIPs throughout the focal archipelagos, including species not sampled in our trees, and then separately analyzed Eurasian-origin species only (no Australo-Papuan-origin species have populations with short-distance migration). Comparisons of medians were made using Mann-Whitney U-tests.

## Competitive pressure in the lowlands

We determined whether individual MIPs shared islands with breeding species from the same genus and family (respectively) occurring in the lowlands (i.e., elevational range extends below 100 masl). Genus- and family-level taxonomic classification follows IOC v 9.2[57], but we treated *Lichmera lombokia* and *Melidectes whitemanensis* (Meliphagidae) as monotypic genera based on the results of Marki et al.[72]. We compared the respective competitive pressure faced by Eurasian-origin versus Australo-Papuan MIPs using chi-squared tests. We then tested whether any differences in potential competitive pressure from lowland relatives could be related to differences in diffuse competitive pressure. To do this, we counted the total number of lowland passerine species sharing islands with every MIP, and compared this for Eurasian-origin versus Australo-Papuan MIPs using a Mann-Whitney U-test.

## Phylogenetic null models

We created a single phylogenetic tree that included 80 species with MIPs (see Methods: Phylogenetic trees for ancestral state reconstructions). We used a well-resolved, dated ultraconserved element (UCE) phylogenetic tree of passerine families by Oliveros et al.[44] as a backbone to represent the interrelationships among groups. We first pruned the tips of the Oliveros et al. consensus tree so that only the focal clades represented in our study remained. Next, we pruned down the clade-level trees generated for our own study to include only species with MIPs (one individual per species). We then grafted these clade-level trees onto the Oliveros et al. backbone. We assessed how well the empirical patterns of the tests above could be replicated based on data simulated under a Brownian motion model of evolution in this tree. To generate null datasets, we first tested the phylogenetic signal in the analysis variables described above using Pagel's $\lambda$[73], also quantifying the $\sigma^2$ values using fitContinuous function in the R package geiger (v.2.0[74]). The maximum likelihood value of $\lambda$ was used to transform the species-level tree using the transformPhylo function in motmot (v.2.1.3[75]). Next, using the $\sigma^2$ values for each variable, we simulated 1,000 null species-level datasets on the transformed trees, using the fastBM function in phytools (v.0.7-70[76]). We then performed one-way ANOVA tests on the empirical data and null datasets, and compared the empirical F statistics against the distributions of the simulated F statistics to assess the divergence in the trait values among groupings. *P*-values were calculated by determining the number of simulated F statistics higher than the empirical F statistic, and dividing this value by the total number of simulations (1,000).

## Taxon sampling of montane supercolonizers

Our sampling provided broad geographic coverage for each montane supercolonizer species/clade, while complementing the taxonomic coverage of sequence data already available from GenBank. We sought sequence data from every subspecies of each species/clade, as well as major geographically disjunct populations within those subspecies. This approach allowed us to perform multi-gene phylogenetic analyses with near complete taxonomic coverage, as well as separate, more powerful genomic analyses drawing exclusively on our own data. Detailed summary statistics on taxonomic sampling are presented in the following sections and in Supplementary Table 6.

In addition to the montane supercolonizers, we sampled eight further individual birds (Supplementary Data 6) from three families to expand the trees used for ancestral state reconstructions (see Methods: Phylogenetic trees for ancestral state reconstructions).

The Indo-Pacific leaf warblers represent a large species complex distributed from the Greater Sundas and the Philippines through Wallacea; across New Guinea and some outlying islands; and through the Bismarcks and Solomons. All populations are allopatric except for

Kolombangara Leaf Warbler *P. amoenus* and Island Leaf Warbler *P. maforensis pallescens*, which co-occur on Kolombangara in the Solomons[59]. Populations are essentially sedentary and montane apart from a few populations that reach the lowlands on small islands. The complex is probably monophyletic, but a comprehensive phylogenetic analysis is lacking (though see Jones & Kennedy[77]; Alström et al.[32]; Ng et al.[78]; Rheindt et al.[35]). Species limits within the group are unclear, and treatment by different taxonomic authorities varies significantly. IOC 9.2[57] recognizes eight species in this complex: *P. trivirgatus* (4 subspecies), *P. nigrorum* (7 subspecies), *P. presbytes* (2 subspecies), *P. rotiensis* (monotypic), *P. makirensis* (monotypic), *P. sarasinorum* (2 subspecies), *P. amoenus* (monotypic), and *P. maforensis* (18 subspecies). Another two taxa were recently described from the Wallacean islands of Peleng and Taliabu[35]. Of these, we sampled 6 of 8 species and 25 of 35 subspecies; additional GenBank data increased coverage to 7 of 8 species, and 100% of subspecies. Missing is the monotypic Rote Leaf Warbler *Phylloscopus rotiensis* from the Lesser Sundas.

Snowy-browed Flycatcher occurs from the Himalayas west of Nepal through southern China and Taiwan; and across Indochina, the Greater Sundas, and Wallacea. Populations in the Philippines have recently been shown not to belong to this species[33]. It has a uniformly montane breeding distribution, but is an altitudinal migrant in the Himalayas. Fourteen subspecies are recognized, including a newly described taxon from Taliabu[35]. We sampled 12 individuals representing 9 of 14 subspecies; supplementary GenBank data increased coverage to 13 of 14 subspecies, missing only subspecies *mjobergi* from the Pueh Mountains of western Borneo. We additionally sampled Damar Flycatcher *Ficedula henrici*, endemic to a single small island in the Lesser Sundas, which bears morphological and vocal similarities to *F. hyperythra*, but has not previously been included in any molecular phylogenetic study.

Mountain Tailorbird is not a true tailorbird (genus *Orthotomus*, family Cisticolidae); rather, it is part of the family Cettiidae, in the genus *Phyllergates*[31], which it shares with one other species from Mindanao. It occurs from the eastern Himalayas through southern China, and Indochina; across the Greater Sundas and Wallacea; and on Palawan and Luzon in the Philippines. Its breeding distribution is exclusively montane, but Himalayan populations are altitudinal migrants. Sixteen subspecies are recognized, including two newly described taxa from Peleng and Taliabu[35]. We sampled 15 individuals representing 11 of 16 subspecies; additional GenBank data increased subspecies coverage to 100%.

### Library preparation and sequencing

We used Illumina sequencing to generate genomic data for the population studies of the three montane supercolonizers. We sequenced *ND2* from eight additional individuals in order to expand the taxonomic coverage of the ancestral state reconstructions. Raw reads have been deposited at the NCBI Sequence Read Archive (SRA). Individual nuclear and mitochondrial genes have been deposited on GenBank. Accession numbers are given in Supplementary Data 6.

For Illumina sequencing, genomic DNA was extracted both from footpad samples ($n = 51$) and from fresh blood and tissue samples ($n = 18$). Protocol for extracting DNA from footpad samples followed Irestedt et al.[79]. To create sequencing libraries suitable for Illumina sequencing of footpad DNA extracts, we followed the protocol of Meyer and Kircher[80]. In short, library preparation consisted of blunt-end repair, adapter ligation, and adapter fill-in, followed by four independent index PCRs. The libraries were run on half a lane on Illumina HiSeq X (pooled at equal ratio with other museum samples). For fresh samples, genomic DNA was extracted with KingFisher Duo magnetic particle processor (ThermoFisher Scientific) using the KingFisher Cell and Tissue DNA Kit. Library preparation (using Illumina

TruSeq DNA Library Preparation Kit) and sequencing on Illumina HiSeqX (2 × 151 bp) was performed by SciLifeLab.

For seven fresh tissue samples, *ND2* (1,041 bp) was sequenced in a single fragment. For a single footpad sample (from *Pachycephala johni* AMNH 658812), *ND2* was sequenced in seven overlapping fragments of <200 bp (Supplementary Data 7). Sequences were assembled using Sequencher v.4.7, and checked for stop codons or indels that would have disrupted the reading frame, and indicated amplification of pseudogenes.

### Bioinformatics

Illumina sequencing reads were cleaned using a custom-designed workflow (available at https://github.com/mozesblom/NGSdata_tools) to remove adapter contamination, low-quality bases and low-complexity reads. Overlapping read pairs were merged using PEAR (v.0.9.10[81]), and Super Deduper (v.1.4[82]) was used to remove PCR duplicates. Trimming and adapter removal was done with TRIMMO-MATIC (v.0.32[83]; default settings). Overall quality and length distribution of sequence reads was inspected with FASTQC (v.0.11.5[84]), both before and after cleaning.

Mitochondrial genomes were assembled using an iterative baiting and mapping approach (MITObim v.1.8[85]; default settings). MITObim locates initial regions of similarity between a target library and a distant reference (reference sequences are listed in Supplementary Table 7). It then employs an iterative mapping strategy to locate reads overlapping with these initial segments, without using the initial reference. Resulting assemblies were corrected and validated by mapping all sequence reads against the inferred assembly using BWA mem[86] (default settings), and checked for remaining variants and major coverage differences that might suggest MITObim had chain-linked a non-mitogenome region. We utilized the complete reconstructed mitochondrial genomes, but also extracted individual mitochondrial genes for certain analyses.

Individual nuclear loci were recovered from the cleaned reads by mapping directly against reference genes (see Supplementary Table 7). Reference sequences were indexed using BWA (v.0.7.12[86]) and SAMtools (v.0.1.19[87]). Consensus sequences were calculated using ANGSD (v.0.933[88]).

In order to compare homologous regions in a phylogenomic analysis, we relied on reference data published by Jarvis et al.[89]. The data are from a broad selection of bird species and consists of individual multiple-sequence alignments from ~19,000 single-copy genes. The Jarvis et al. data were filtered on length, and we retained 17,341 alignments with a length of 200–5,000 bp. Our filtered Illumina data was first assembled with Megahit (v.1.2.8[90]), using default settings. Next, homologous regions corresponding to the multiple sequence alignments in the reference data were located and extracted from the genome assemblies. This was done using HMMer (v.3.2.1[91]), as implemented in the BirdScanner workflow (https://github.com/nylander/birdscanner2).

### Phylogenetic analyses of montane supercolonizers

We built supermatrix trees[92] for each of the three montane super-colonizers, complementing existing sequence data from GenBank with our own sampling (Supplementary Data 6). We used as a foundation three phylogenetic datasets produced for the ancestral state reconstructions: those for *Phylloscopus* (Phylloscopidae), *Ficedula* (Musci-capidae), and Cettiidae (see Methods: Phylogenetic trees for ancestral state reconstructions). We retained the gene sets from those datasets (5–7 nuclear and mitochondrial genes), but sampled many more individuals and taxa from the groups of interest. Note that the *Phylloscopus* tree used for ancestral state reconstructions included all available individuals and taxa, so it was reused without further additions. Phylogenetic analyses were performed as described in Methods:

Phylogenetic trees for ancestral state reconstructions. We used linked substitution models for mitochondrial genes: GTR + I + G for *Phylloscopus*, and GTR for *Ficedula* and Cettiidae.

We also performed phylogenetic analyses on the newly generated mitochondrial genome data from each of the three montane super-colonizers. Analyses were performed as described in Methods: Phylogenetic trees for ancestral state reconstructions. To obtain absolute date estimates, we partitioned the *cyt-b* and *ND2* data, and applied substitution rates from Lerner et al.[66]. We used the GTR nucleotide substitution model. Outgroups were *Phylloscopus claudiae* for the Indo-Pacific *Phylloscopus* analysis; *Ficedula zanthopygia* for the *Ficedula hyperythra* analysis; and *Horornis parens* for the *Phyllergates cucullatus* analysis, based on Alström et al.[32], Moyle et al.[33], and Alström et al.[31], respectively.

Phylogenomic analyses were performed by extracting putative homologous gene regions from all genome assemblies, and aligning these individually using MAFFT (v.7.310[93]) with the '--auto' option for automatic algorithm selection, followed by a filtering step using BMGE (v.1.12[94]), where uncertain alignment regions are removed. The *Phyllergates cucullatus meisei* individual was removed as it showed a high proportion of missing data. Gene trees were then inferred using RAxML-NG (v.0.9.0[95]) using the GTR + G model. Alignments were then further filtered by identifying and removing long-branch taxa using TreeShrink (v.1.3.3[96]), followed by a final round of MAFFT, BMGE, and RAxML-NG. The final set of gene trees were analyzed in ASTRAL III (v.5.6.3[97]) to produce an estimate of a species tree. In addition, all individual gene alignments were concatenated (a total of 9,295,039 bp) and a phylogeny was estimated with RAxML-NG using the GTRI + I + G model. The simultaneous inference of thousands of gene trees were facilitated by using the ParGenes v.1.0.1 workflow[98], and using the GNU parallel (v.20210822[99]) library. Complete analysis workflow is available from GitHub (https://github.com/nylander/birdscanner2).

### Reporting summary

Further information on research design is available in the Nature Portfolio Reporting Summary linked to this article.

## Data availability

Raw Illumina sequences are deposited in the Sequence Reads Archive, National Center for Biotechnology Information [https://www.ncbi.nlm.nih.gov/bioproject/PRJNA747888]. Individual gene sequences are deposited on GenBank, accession numbers OM991446–OM991474, ON015307–ON015422, and ON015448–ON015647. Additonal source data are presented in Supplementary Data 1–7. Source data for the Figures can be found in Supplementary Data 1 and 3.

## Code availability

Code and workflow for BirdScanner, used to extract and compare homologous gene regions, is available at https://github.com/nylander/birdscanner2.

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

## Acknowledgements

Genetic samples were kindly provided by the following institutions: the American Museum of Natural History, New York, NY (Paul Sweet, Tom Trombone, Brian Tilston Smith, and Peter Capainolo); the Australian National Wildlife Collection (Leo Joseph and Robert Palmer); the British Museum of Natural History, Tring (Robert Prys-Jones, Hein van Grouw, Alexander L. Bond, and Mark Adams); Museum für Naturkunde, Berlin (Sylke Frahnert and Pascal Eckhoff); the Natural History Museum of Denmark (Jan Bolding Kristensen); Rijksmuseum van Natuurlijke Histoire, Leiden (Steven van der Mije and Pepijn Kamminga); the Swedish Museum of Natural History, Stockholm (Ulf Johansson); and the Yale Peabody Museum of Natural History, New Haven, CT (Kristof Zyskowski). We thank Petter Marki and Michael Le Pepke for contributing data. Leo Joseph, Frederick Sheldon, Trevor Price, and Jon Fjeldså provided valuable comments on the draft manuscript. We acknowledge support from the Villum Foundation (Young Investigator Programme, project No. 15560, K.A.J), the Carlsberg Foundation (CF15-0078 and CF15-0079, K.A.J.; CF17-0239, J.D.K), Marie Sklodowska-Curie actions (MSCA-792534, J.D.K.), and the Swedish Research Council (2019-04486, P.A.). The computations were performed on resources provided by SNIC through Uppsala Multidisciplinary Center for Advanced Computational Science (UPPMAX) under project SNIC 2017/7-212. The authors acknowledge support from the National Genomics Infrastructure in Stockholm, funded by Science for Life Laboratory, the Knut and Alice Wallenberg Foundation, and the Swedish Research Council. We also thank SNIC/Uppsala Multidisciplinary Center for Advanced Computational Science for assistance with massively parallel sequencing, and access to the UPPMAX computational infrastructure.

## Author contributions

A.H.R. and K.A.J. conceived the study. A.H.R., J.D.K., J.M.P., B.P., M.P.K.B. P.A., T.H., P.G.P.E., M.I., J.A.A.N. and K.A.J. contributed to build the dataset. A.H.R., J.D.K., J.A.A.N. and K.A.J. developed the analytical framework. J.M.P., B.P., M.P.K.B., P.G.P.E. and J.A.A.N. performed bioinformatics. A.H.R., J.D.K., J.A.A.N. and K.A.J. performed the analyses. A.H.R. led the writing, and A.H.R., J.D.K., J.M.P., B.P., M.P.K.B., P.A., T.H., P.G.P.E., M.I., J.A.A.N. and K.A.J. contributed to the discussion of the results and the writing of the manuscript.

## Competing interests

The authors declare no competing interests.
