## [Peer Review File · Nature Communications]

Reviewers' Comments:

Reviewer #1:

Remarks to the Author:

This is a quite interesting study examining the historical assembly of montane bird communities in Wallacea and the Bismarck and Solomon archipelagos from a phylogenetic perspective. The main conclusion is that lineages with an Eurasian origin extended via repeated colonization between mountains, whereas those with an Australo-Papuan origin evolved from lowland ancestors which, once colonizing highlands did not further disperse to other island mountains. The data sets are comprehensive and well analyzed and the study has potential to be an important contribution to biogeography. My main concern, however, is the lack of a comprehensive theoretical framework with well-delineated hypotheses and predictions in the Introduction. I elaborate on this issue in my specific comments below.

82-91: You may wish to review classic work by Chapman (1917, 1926) and later by Vuilleumier in the Andes, who originally proposed that cool montane areas of the Neotropics were colonized by birds from temperate areas in southern South America and not from the adjacent lowlands. This same idea was elaborated on in several studies of plants, e.g. see Hughes & Eastwood (2006, PNAS).

90: Migration might in fact be an adaptation to escape seasonal climates (e.g. see cases of migrants tracking their climatic niches in space; <http://dx.doi.org/10.1098/rspb.2015.2458>), so it would be good to elaborate what exactly would be the mechanism whereby migrants are expected to be better able to colonize mountains on islands – is it because of their putative ability to tolerate diverse climates as implied here or instead because they move around more and hence are more prone to vagrancy and establishment at new locations?

91: More specific predictions or further elaboration on the patterns being tested would be helpful here. Several of the results appear somewhat as surprises because they cannot be traced directly to the one hypothesis described here. The paper does quite a bit more than test this particular hypothesis and this needs to be clear so that readers clearly understand the significance of all results given the theoretical framework of the Introduction. For example, it should be clear that (and why) examining the proportion of MIPs relative to total island populations is a way to gauge support for the taxon cycle hypothesis. Also, results describe analyses aimed to examine potential for competition, but this comes up totally as a surprise because no role for competition in the assembly of montane biotas was mentioned at all in the Introduction. I realize that some of the explanations appear later on (in the Methods), but I would strongly recommend that all the angles that the study explores are clearly laid out with appropriate context in the Introduction.

97: “novel bioinformatic methods” is vague – please briefly describe what is it that the method does more precisely.

98: what do you mean by broadest? Most specious?

103-114: how were these clades identified/defined?

119-123: briefly define what do you mean by “ancestral source nodes”.

129-131: Why this is relevant/interesting depends on how the clades were defined.

182-190: why report mean values but conduct statistical tests based on median values?

193-194: how do you gauge “potential from competition from lowland congeners” and how can one know whether competition in the lowlands might drive MIP formation? And why is this something

being addressed in the context of the framework outlined in the Introduction? I know this is all better explained in the methods further down, but it is important to bear in mind that given the Nat Comms style, readers will arrive at results first, so it is important that this section is fully self explanatory.

195-196: why consider only competition from species in the same family and not diffuse competition with the entire assemblage? Is there any relationship with the total number of species in the lowlands (i.e. not only those in the same family)?

202-219: This is confusing. If there is strong phylogenetic signal in the data, then why don't you explicitly account for phylogeny using appropriate comparative methods? In the absence of such a correction, it is impossible to assess which of the putative effects might be real and independent from phylogeny as opposed to the study simply being a comparison of two clades which happen to differ in multiple correlated attributes including their geographic range.

221: What exactly is the point of these analyses? Again, I understand this might be well explained in Methods, but at the very least the specific questions or hypotheses being addressed need to be clearly stated. Also, focus on the results that link directly to the main objectives of the study and exclude details that might be crucial for other kind of work (e.g. that taxonomy is inconsistent with phylogeny) but that here do not seem necessary and cause distraction. More broadly, I find that the detailed phylogenetic analyses of a few groups are not well integrated with the broader analyses in the paper. Perhaps the relationship between these different datasets/analyses could be better articulated if the authors were more clear about their theoretical framework and all the questions/hypotheses/predictions were better outlined in the Introduction (perhaps a conceptual figure would help?), but I wonder if another possibility would be to drop the analyses on individual groups from the current manuscript and publish these separately, with more space to discuss data and results in two separate manuscripts.

268-269: see also Moyle et al 2017 10.7717/peerj.3335.

275: Can you please explain precisely which of the results indicated that mountain colonization is a dead-end for further colonizations of mountains in Australo-Papuan lineages?

I was left wondering whether in this system you see within-mountain speciation. In areas like the Himalayas there has been little to no diversification within the mountain system, with diversity accumulating largely as a result of colonization by lineages that diversified elsewhere, but in areas such as the Andes, there are large montane radiations where ancestors which were seemingly widespread diversified into daughter lineages occupying separate mountains.

286-288: Again, I do not quite understand the idea of dead-ends as used here (vs. stepping stone).

Figures 3 and 5: It is not clear how were these plots showing distributions and (presumably) median values constructed – what are the individual data points used to plot such distributions? More information is required in the legends.

311: what is "regional evolution"?

314: Why competition is seemingly ruled out is not entirely clear, largely because the authors did not set up their hypothesis and predictions in a proper conceptual framework in the Introduction. See e.g. DOI: 10.1126/science.abl7242

400: having $n=1$ from each region, I would be careful to establish such temperate/tropical dichotomy as a main conclusion.

Figure 6: nice illustrations, but they do not add valuable information regarding results of the study.

Perhaps use these nice images in a composite figure showing results of analyses?

Reviewer #2:

Remarks to the Author:

This manuscript presents the results of a study of the processes that have produced patterns in the distribution of birds in montane areas within the Indo-Pacific region. The sampling of taxa is generally excellent, the genetic data are extensive, the analyses are extensive and detailed, and the conceptual context is solid. The difference in the long-term dynamics of colonization and diversification between their two primary study areas is clearly demonstrated, and rightly deserves strong emphasis: not all birds follow the same patterns of biogeographical dynamics. The analytical aspects of the manuscript are especially strong. The paper will be of interest to a broad set of people who study avian biogeography; the dynamics of avian island community composition; the biogeography of the Indo-Pacific region; evolutionary biology of phylogeny and speciation processes; and conservation biologists who study persistence and extinction.

Because the study has been conducted in a thorough fashion, and the paper is well organized and is clearly written, my suggestions are relatively few and fairly minor, and mostly involve expanding some issues already noted. In no particular order, they are the following.

1. The keywords seem sparse. Consider adding some of these: "endemism, extinction, persistence, speciation, taxon cycles, Wallacea, Bismarcks, Solomons".

2. The term "Wallacea" has been used variably since being coined, and needs some definition here. When proposed by Merrill (1924) and described in greater detail by Dickerson et al. (1928), it had the Philippines as one of two core areas, the second being the southern region that the authors of this paper refer to as Wallacea. The authors' usage is fairly common currently but is not universal, and some recent analyses have presented evidence supporting the original definition; for a summary of the long history of the term, see Ali and Heaney (2021, 2022). I suggest the authors make it explicit that they are using the term for what is mostly the Indonesian portion of Wallacea, following the primary research paper of their choice. Using a popular-level field guide to birds for their use of the term is probably not the best idea. To be clear, I don't object to their decision to limit their study in the way they have; they just need to be more careful with their use of terminology.

3. Similarly, the term "taxon cycles" has been used principally in two somewhat different ways. The authors point this out well into the paper (line 338 and following), but do so rather briefly. Since this model is an important part of their paper, it seems they should at least briefly note in the Introduction (lines 76-78) that they follow the definition and description of Ricklefs and Cox (1978) and Ricklefs and Bermingham (2002), rather than the original formulation by Wilson (1961). The primary difference is that Wilson emphasized that the pattern of colonization-speciation-competition-upslope migration-extinction was repeated many times, and that it is the repetition that gives rise to the notion of a cycle. Ricklefs mostly dropped the repetitiveness of the complete cycle, and emphasized repeated colonization, speciation, and upslope movement. What the authors have found in this study is indeed closest to what Ricklefs described, so any changes simply need to clarify what they mean (and do not mean) by "taxon cycle".

4. A majority of the studies that are relevant to this paper have focused on birds, and so it is appropriate that the authors emphasize that literature. However, they do so rather exclusively, without noting that their summaries of the literature emphasize birds. I suggest that they insert the word "avian" or "birds" appropriately to indicate this. I suggest that they also cite at least a few of the papers on other taxa (mammals, herps, insects) that are relevant to this study. It may be the case that few of these papers include "taxon cycle" in their titles, but that does not mean that they are not relevant.

5. The authors show clear evidence that species of birds that have colonized montane areas widely within Wallacea (as they define it) are related to continental species that are migratory, and they view migratory behavior as an important pre-adaptation. That is clearly demonstrated by their data. However, they briefly touch on another aspect that they reject without due consideration, physiological adaptation to the cool climates that characterize montane habitats in the tropics. A knowledgeable colleague tells me that many of the birds on the Eurasian continent that are migratory have been studied and demonstrate physiological features that allow them to deal with cool/cold conditions while also retaining the ability to deal with hot conditions. This suggests to me that physiological plasticity and migratory behavior may both contribute to the success of these colonizing birds, and may even be parts of a tightly co-evolved set of adaptations. Currently, this hypothesis is given short shrift.

6. A bit further afield is a topic that might deserve explicit mention. The calibrated phylogenies presented here show that many of these taxa evolved in the early Pleistocene, and some earlier. This means that they have persisted through many (perhaps as many as 24) major climatic (and sea level) fluctuations. This level of persistence seems notable in its own right; it suggests that persistence is an important part of this system, and that extinction can have had less than overwhelming effects; stated another way, turnover is less apparent than persistence. That topic may deserve some development.

Reviewer #3:

Remarks to the Author:

Dear Dr. Ward,

In the article "The formation of the Indo-Pacific montane avifauna" the authors try to understand the processes behind the formation of the avian diversity in mountain ranges in islands in Southeast Asia and part of Oceania. They use phylogenetic trees and ancestral range reconstruction to access the origin of 80 bird species complexes spread across their focal areas in Wallacea, Birmarcks and Solomons. In general, the article is well written, as the authors do a good job at contextualizing their goals and achievements in the existing theoretical framework. However, I have some concerns that should be addressed before I can recommend this paper for acceptance at Nature Communications:

1) The article's main flaws are in the dataset used to build most of their phylogenetic trees, which are a handful of nuclear and mitochondrial genes. Especially for applying BioGeoBEARS, I would expect phylogenetic trees built from reduced representation genomic data at the species/subspecies level. I am not proposing the authors go back and re-do their entire work, but they need to acknowledge the fact that their trees are just a hypothesis of branching patterns and so are the conclusions regarding ancestral ranges. It is also important to take in consideration the magnitude in which trees produced from mitochondrial markers are discordant in relation to trees from nuclear markers.

2) The paper has two major analytical approaches, one at the species level and one at the populational level for a few focal groups. Although the first part is framed in a hypothesis testing structure, the second part seems disconnected and needs to be better integrated with the paper main goal. There should be more information on how the focal species were selected and how their selection could bias the paper conclusions in one specific direction.

3) In general, the language of the paper treats ancestral reconstructions as the "true past". The article language should reflect the fact that ancestral reconstructions are only hypotheses of past range and behaviors.

4) The authors need to be especially careful with their results on competitive pressures, as they are using the shortcut of presence/absence of lineages in the same genus/family as evidence for competitive pressures between bird species. This a fragile assumption, especially considering that

outcompeting can lead to extinction, and the methods applied here are practically blind to extinct lineages.

a few specific comments:

line 178: weird use of "overwhelmingly", probably replace with "mostly"

line 204-205: please explain what F being greater in empirical datasets than in simulated data means in the context of your hypothesis testing.

line 274: please rephrase acknowledging the limitations of your results.

Line 624: this section needs to be better explained. How was Oliveros' tree reconciled with your trees?

REVIEWER COMMENTS

Reviewer #1 (Remarks to the Author):

This is a quite interesting study examining the historical assembly of montane bird communities in Wallacea and the Bismarck and Solomon archipelagos from a phylogenetic perspective. The main conclusion is that lineages with an Eurasian origin extended via repeated colonization between mountains, whereas those with an Australo-Papuan origin evolved from lowland ancestors which, once colonizing highlands did not further disperse to other island mountains. The data sets are comprehensive and well analyzed and the study has potential to be an important contribution to biogeography. My main concern, however, is the lack of a comprehensive theoretical framework with well-delineated hypotheses and predictions in the Introduction. I elaborate on this issue in my specific comments below.

RESPONSE:

We very much appreciate the reviewer's detailed and positive evaluation of our work and the resulting suggestions for improvement. We have made a considerable effort to clarify the hypotheses and theoretical framework of the manuscript by reorganizing the text, providing necessary context, and editing and expanding passages that were previously unclear. We have carefully addressed all of the individual comments, which we believe has resulted in a much stronger manuscript. Point-by-point responses are given below.

82-91: You may wish to review classic work by Chapman (1917, 1926) and later by Vuilleumier in the Andes, who originally proposed that cool montane areas of the Neotropics were colonized by birds from temperate areas in southern South America and not from the adjacent lowlands. This same idea was elaborated on in several studies of plants, e.g. see Hughes & Eastwood (2006, PNAS).

RESPONSE:

Thanks very much for bringing these to our attention. We have added citations to the 1917 and 1926 works by Chapman (lines 95, 237, and 417), and to the 2006 Hughes & Eastwood paper (line 86).

90: Migration might in fact be an adaptation to escape seasonal climates (e.g. see cases of migrants tracking their climatic niches in space; <http://dx.doi.org/10.1098/rspb.2015.2458>), so it would be good to elaborate what exactly would be the mechanism whereby migrants are expected to be better able to colonize mountains on islands – is it because of their putative ability to tolerate diverse climates as implied here or instead because they move around more and hence are more prone to vagrancy and establishment at new locations?

RESPONSE:

Beyond the general connection with increased mobility, we do not have any firm expectations about what the precise mechanism might be. We do not intend to imply that an ability to tolerate diverse climates leads to increased success in colonizing island mountains. Our study shows that short-distance (as opposed to long-distance) migration is strongly associated with species' success in colonizing island mountains. We have added a sentence to the Discussion (lines 435–438) speculating that “Short-distance migration may be linked with an ability to rapidly and flexibly track shifting resources, facilitating colonization of island mountains; however,

relationships between niche breadth, dispersal capacity, and range size in short-distance migrants are not well understood (Laube et al., 2015; Gómez et al., 2016).”

91: More specific predictions or further elaboration on the patterns being tested would be helpful here. Several of the results appear somewhat as surprises because they cannot be traced directly to the one hypothesis described here. The paper does quite a bit more than test this particular hypothesis and this needs to be clear so that readers clearly understand the significance of all results given the theoretical framework of the Introduction. For example, it should be clear that (and why) examining the proportion of MIPs relative to total island populations is a way to gauge support for the taxon cycle hypothesis. Also, results describe analyses aimed to examine potential for competition, but this comes up totally as a surprise because no role for competition in the assembly of montane biotas was mentioned at all in the Introduction. I realize that some of the explanations appear later on (in the Methods), but I would strongly recommend that all the angles that the study explores are clearly laid out with appropriate context in the Introduction.

RESPONSE:

Based on this and many of the other reviewer comments, we recognize that the organization of the Methods and Results sections in the previously submitted version of the manuscript was confusing to readers because the results were presented without proper context. We have heavily restructured these sections by moving explanations of rationale and hypotheses for the different analyses from the Methods to the Results wherever we could see that the results were not readily interpretable. We fully agree with Reviewer #1’s observation (below) that, because readers will arrive at the Results first, this section must provide enough context to be fully self-explanatory.

The reviewer has a valid concern that we do not clearly outline all angles of the study in the Introduction. We have added the following sentences to the last paragraph of that section (lines 108–113): “We conducted additional analyses to evaluate whether species’ modern distributions are compatible with taxon cycles, how different migratory behaviors correlate with species’ success in colonizing island mountains, and whether competitive pressure from close relatives drives montane distributions. Our results provide evidence that the establishment of montane island populations follows fundamentally different processes depending on the long-term region-specific evolution of the parent lineage.” We have refrained from elaborating on the rationale and specific hypotheses for these additional analyses in the Introduction in order to keep this section relatively concise, and so as not to distract from the main narrative of the Introduction and core hypotheses of the study. Necessary context for all analyses is now provided in the heavily revised Results section. For example, we describe in lines 193–203 how examining the proportion of MIPs relative to total island populations is a way to gauge support for the taxon cycle hypothesis.

97: “novel bioinformatic methods” is vague – please briefly describe what is it that the method does more precisely.

RESPONSE:

We have changed the wording from “a novel bioinformatics method” to “a new method to extract and compare homologous gene regions” (lines 105–106).

98: what do you mean by broadest? Most specious?

RESPONSE:

We mean geographically broadest. The sentence has been edited accordingly (lines 104–108).

103-114: how were these clades identified/defined?

RESPONSE:

We moved the following passage from the Methods to the Results (lines 124–127): “Clade limits were defined with the goal of determining Eurasian or Australo-Papuan origin (or lack thereof) for species with MIPs. The clades as defined were approximately 5–15 Ma old. Each encompassed no more than one family.”

119-123: briefly define what do you mean by “ancestral source nodes”.

RESPONSE:

We have edited the paragraph in question (lines 134–138). The first two sentences now read: “We defined the geographic origin of the 80 focal species using ancestral ranges inferred with BioGeoBEARS (Matzke 2013a, 2013b). We counted back from terminal tree nodes until reaching “Ancestral Source Nodes” with > 75% probability of being either Eurasian (Palearctic + Indomalaya) or Australo-Papuan (see Methods: Ancestral state reconstructions).”

129-131: Why this is relevant/interesting depends on how the clades were defined.

RESPONSE:

Please see our response to Reviewer 1’s comment on lines 103-114. We moved the following passage from the Methods to the Results (lines 124–127): “Clade limits were defined with the goal of determining Eurasian or Australo-Papuan origin (or lack thereof) for species with MIPs. The clades as defined were approximately 5–15 Ma old. Each encompassed no more than one family.”

182-190: why report mean values but conduct statistical tests based on median values?

RESPONSE:

We now report median values rather than mean values in line with our statistical tests. The following sections are affected:

- Results: Montane vs. lowland ancestry (lines 162–167)
- Results: MIPs and lowland island populations (LIPs) (lines 192–233)
- Results: Migration (lines 235–265)
- Table 2

193-194: how do you gauge “potential from competition from lowland congeners” and how can one know whether competition in the lowlands might drive MIP formation? And why is this something being addressed in the context of the framework outlined in the Introduction? I know this is all better explained in the methods further down, but it is important to bear in mind that given the Nat Comms style, readers will arrive at results first, so it is important that this section is fully self explanatory.

RESPONSE:

We have moved some relevant text from the Methods to the Results as part of our overall restructuring of these sections. We have also expanded on and clarified our reasoning to show how the question of competitive pressure fits into the framework outlined the Introduction. The new passage (lines 268–274) in the Results reads: “Competitive pressure from closely related lowland species is often implicated in the formation of new montane populations from lowland

source pools (Terborgh, 1971). This mechanism may not be important for species colonizing directly between mountains and retaining their elevational niches. In line with these hypotheses, Australo-Papuan-origin MIPs often share islands with lowland relatives while Eurasian-origin MIPs do not. To quantify this pattern, we determined whether individual MIPs share islands with breeding species from the same genus and family (respectively) occurring in the lowlands.”

We have also added the following note about for interpretation of these results (lines 284–286): “We stress that the results presented in this section indicate the *potential* for competition from lowland relatives. Direct evidence requires field experiments (Jankowski et al., 2010; Freeman et al., 2016).”

195-196: why consider only competition from species in the same family and not diffuse competition with the entire assemblage? Is there any relationship with the total number of species in the lowlands (i.e. not only those in the same family)?

RESPONSE:

We tested for this, counting all lowland passerine species sharing islands with each individual MIP, and comparing Eurasian- vs. Australo-Papuan-origin MIPs. We found no significant difference in diffuse competition, but we agree that including this analysis provides useful context for the results already reported in the previous version. Accordingly, we have made additions to the Results section (lines 280–284) and the Methods section (lines 783–787), and added the data to Supplementary Data 1.

202-219: This is confusing. If there is strong phylogenetic signal in the data, then why don't you explicitly account for phylogeny using appropriate comparative methods? In the absence of such a correction, it is impossible to assess which of the putative effects might be real and independent from phylogeny as opposed to the study simply being a comparison of two clades which happen to differ in multiple correlated attributes including their geographic range.

RESPONSE:

We fully accept the need to assess whether the patterns we observe are independent from what we would expect as a consequence of phylogenetic relatedness alone. We believe this has already been achieved by the methodological approach presented in the previous submission. Where the F-values derived from our Brownian motion simulations are significantly lower than those observed in the empirical data, these effects can be considered greater than expected based on phylogenetic relatedness alone. We employed Brownian motion simulations because of the nature of our data (probabilities of character states, numbers of populations) and the heavily skewed distributions of that data. These data attributes necessitated using parametric Mann-Whitney U-tests to assess significant differences; we are not aware of an analogous phylogenetic version of this test. We note that Reviewer 3 also found this section confusing (see their comment on lines 204-205 of the previous version). In the revised version of the manuscript, we have tried to better clarify our approach to testing for phylogenetic signal and the value of this method (lines 289–307).

A minor point is that the study deals with three major clades (Passerides, Meliphagides, and Corvides). The relevant passage in the previous version could easily be interpreted as describing two clades – we have edited it accordingly (lines 316–319).

221: What exactly is the point of these analyses? Again, I understand this might be well explained in Methods, but at the very least the specific questions or hypotheses being addressed

need to be clearly stated. Also, focus on the results that link directly to the main objectives of the study and exclude details that might be crucial for other kind of work (e.g. that taxonomy is inconsistent with phylogeny) but that here do not seem necessary and cause distraction. More broadly, I find that the detailed phylogenetic analyses of a few groups are not well integrated with the broader analyses in the paper. Perhaps the relationship between these different datasets/analyses could be better articulated if the authors were more clear about their theoretical framework and all the questions/hypotheses/predictions were better outlined in the Introduction (perhaps a conceptual figure would help?), but I wonder if another possibility would be to drop the analyses on individual groups from the current manuscript and publish these separately, with more space to discuss data and results in two separate manuscripts.

RESPONSE:

We feel that the population studies are a crucial component of our study that allow a much more direct investigation of processes that we can only infer with the other analyses. We show that the most successful mountain colonizers in the region have indeed followed the pathways and processes hypothesized in this study, and that greatly strengthens the overall results. Nevertheless, we agree that the presentation of the population studies was lacking. To address this, we have shifted important information about the hypotheses and background from the Methods to the Results, and heavily edited the latter section to clarify the questions being asked and how they relate to the core aims of the study. The relevant section in the Results now begins with two paragraphs that adequately introduce and contextualize the population studies (lines 322–347).

We have relegated all discussion of taxonomy vs. phylogeny to the Supplementary Information so that it does not distract from the core message. We instead make clear that it is important to establish whether the species/clades analyzed actually represent single radiations. Lines 337–341: “...we wanted to confirm that the species/clades in question are actually monophyletic, representing single radiations. Modern molecular work regularly reveals Indo-Pacific island “species” to be aggregations of similarly-plumaged but unrelated populations (see e.g. Andersen et al., 2014; Moyle et al., 2015).” And on lines 356–358: “The groups analyzed are mostly monophyletic and represent individual island radiations, although the single-island endemic Damar Flycatcher *Ficedula henrici* is embedded within Snowy-browed Flycatcher.”

268-269: see also Moyle et al 2017 10.7717/peerj.3335.

RESPONSE:

Thanks – we have added a citation to this paper on line 397.

275: Can you please explain precisely which of the results indicated that mountain colonization is a dead-end for further colonizations of mountains in Australo-Papuan lineages?

RESPONSE:

We find that Australo-Papuan-origin MIPs evolved from lowland ancestors (Results: Montane vs. lowland ancestry). We infer that ancestral lowland species colonized islands, and in some cases gave rise to MIPs on those islands. Few Australo-Papuan-origin species have multiple MIPs (Results: Establishing multiple MIPs). Species colonizing archipelagos directly between island mountains should tend to have many MIPs and few/no lowland populations, but this pattern is basically absent among Australo-Papuan-origin species (Results: MIPs and LIPs).

We have expanded the relevant passage with a brief summary of our thinking here, with reference to relevant figures and results (lines 407–411): “Conversely, Australo-Papuan-origin

MIPs appear to have arisen via past island colonizations by lowland ancestors (Fig. 2), and little in their modern distribution patterns indicates an ability to colonize directly between the mountains of different islands (Figs. 3 and 4; see also Results: MIPs and LIPs). For these species, mountains represent dead-ends for further dispersal.”

I was left wondering whether in this system you see within-mountain speciation. In areas like the Himalayas there has been little to no diversification within the mountain system, with diversity accumulating largely as a result of colonization by lineages that diversified elsewhere, but in areas such as the Andes, there are large montane radiations where ancestors which were seemingly widespread diversified into daughter lineages occupying separate mountains.

RESPONSE:

There is no clear evidence of parapatric speciation along elevational gradients on the focal islands. Very few montane island populations have sister species replacing them elevationally on the same island. In these few cases, the species pairs either diverged long ago (e.g., *Enodes erythrophris* and *Scissirostrum dubium* on Sulawesi), or occur on multiple islands (e.g., *Dicaeum sanguinolentum* and *D. igniferum* in the Lesser Sundas). Studies from adjacent regions have also failed to find evidence of parapatric speciation along mountains (Greater Sundas: Sheldon et al., 2015; Moyle et al., 2017; New Guinea: Linck et al., 2020; Pujolar et al., 2022).

We don’t find any evidence of montane radiations by Australo-Papuan-origin species in our study. However, the stepping stone colonizations we find for a number of Eurasian-origin species/lineages could certainly be considered “montane radiations.” Most of these radiations are young, but given enough time the individual island populations may eventually speciate, which seems to be happening in the *Phylloscopus* clade we analyzed. Homogenizing gene flow between island populations could potentially hinder this process – several authors of this MS recently explored that question in a population study of *Turdus poliocephalus* (Reeve et al., 2023).

286-288: Again, I do not quite understand the idea of dead-ends as used here (vs. stepping stone).

RESPONSE:

Please see our response to the above comment regarding line 275 of the previous manuscript version.

Figures 3 and 5: It is not clear how were these plots showing distributions and (presumably) median values constructed – what are the individual data points used to plot such distributions? More information is required in the legends.

RESPONSE:

To aid interpretation of the violin plots in Figs. 2, 3, and 5, we added the following sentence to each of the respective captions: “The violin plots show full data distributions superimposed over standard boxplots.” Figs. 3 and 5 show sample sizes below the individual violin plots, and we now make explicit in the captions what these refer to: “The numbers of species included in each sample are given below the respective plots.”

311: what is “regional evolution”?

RESPONSE:

We have slightly modified the language throughout to make this clearer.

In the Introduction (lines 90–95): “Passerine birds have entered the many islands scattered between Eurasia and Australo-Papua from one or the other continental source. The establishment of montane island populations may follow fundamentally different processes depending on the long-term regional-specific evolution of the parent lineage. Lineages from temperate regions may be more successful than tropical lineages at achieving broad and rapid colonization across mountain ranges (Chapman, 1917, 1926; Lobo & Halffter, 2000; Donoghue, 2008; Merckx et al., 2015).”

In the Discussion (lines 377–379): “We find that montane island populations’ (MIPs) modes of colonization, and their potential for further montane colonizations, are governed by the long-term regional-specific evolution of their parent lineages.”

In the Discussion (lines 448–452): “However, as we demonstrate, the ability to colonize directly between mountains is disproportionately manifested among species that come from a specific temperate evolutionary background, and that have behavioral traits reflective of that ancestry (i.e., short-distance migration). This speaks strongly for ~~regional evolution~~ region-specific selective pressures being as the causative mechanism.”

In the Discussion (lines 547–550): “As our knowledge of the phylogenetic relationships of species rapidly improves, it will be increasingly feasible to investigate how the regional-specific evolution of lineages has shaped contemporary global biodiversity patterns.”

314: Why competition is seemingly ruled out is not entirely clear, largely because the authors did not set up their hypothesis and predictions in a proper conceptual framework in the Introduction. See e.g. DOI: [10.1126/science.ab17242](https://doi.org/10.1126/science.ab17242)

RESPONSE:

Please see this reviewer’s above comment on lines 193-194 of the previous manuscript version, and our response to it. We have moved some relevant text from the Methods to the Results as part of our overall restructuring of these sections. We have also expanded on and clarified our reasoning to show how the question of competitive pressure fits into the framework outlined the Introduction. The new passage (lines 268–274) in the Results reads: “Competitive pressure from closely related lowland species is often implicated in the formation of new montane populations from lowland source pools (Terborgh, 1971). This mechanism may not be important for species colonizing directly between mountains and retaining their elevational niches. In line with these hypotheses, Australo-Papuan-origin MIPs often share islands with lowland relatives while Eurasian-origin MIPs do not. To quantify this pattern, we determined whether individual MIPs share islands with breeding species from the same genus and family (respectively) occurring in the lowlands.”

We have also added the following note about for interpretation of these results (lines 284–286): “We stress that the results presented in this section indicate the *potential* for competition from lowland relatives. Direct evidence requires field experiments (Jankowski et al., 2010; Freeman et al., 2016).”

400: having n=1 from each region, I would be careful to establish such temperate/tropical dichotomy as a main conclusion.

RESPONSE:

We have edited the language of this sentence to be slightly more equivocal (lines 546–547): “This dichotomy appears to reflect an important divide between temperate and tropical lineages.” However, beyond the comparison of Eurasian- vs. Australo-Papuan-origin avifaunas, this statement is supported by our examination of the tropical vs. temperate dichotomy within the Eurasian-origin avifauna itself. Lowland Indomalayan (tropical) lineages have not generated any montane island populations in the focal archipelagos (see paragraph 2 of the Discussion). Most Eurasian-origin MIPs had ancestors occurring in the mountains of the (temperate) Palearctic (see Results: Geographic origins of montane island populations). Our findings that most Eurasian-origin species with MIPs have migratory ancestors, and that short-distance migrant species form most MIPs (Results: Migration) further links direct colonization between mountains to temperate and seasonal climates. The population studies of montane supercolonizers also indicate ancestral conditions of temperate occurrence and migratory behavior. Given these different lines of evidence, we think that highlighting a temperate/tropical dichotomy in the last paragraph of the Discussion is warranted.

Figure 6: nice illustrations, but they do not add valuable information regarding results of the study. Perhaps use these nice images in a composite figure showing results of analyses?

RESPONSE:

This is a fair point, but a considerable part of the manuscript is devoted to analysis and discussion of these three species/clades, and we think this will be more compelling to readers if they know what the birds look like. We agree that the presentation of the bird illustrations on their own without additional visual context is slightly awkward, so we integrated them into Figure 1, along with illustrations of two Australo-Papuan species. This serves the dual purpose of visually reinforcing the comparison of the Eurasian- and Australo-Papuan-origin avifaunas, and also giving readers some images to connect with the discussion of “montane supercolonizers.” The Figure 1 caption has been revised accordingly.

Reviewer #2 (Remarks to the Author):

This manuscript presents the results of a study of the processes that have produced patterns in the distribution of birds in montane areas within the Indo-Pacific region. The sampling of taxa is generally excellent, the genetic data are extensive, the analyses are extensive and detailed, and the conceptual context is solid. The difference in the long-term dynamics of colonization and diversification between their two primary study areas is clearly demonstrated, and rightly deserves strong emphasis: not all birds follow the same patterns of biogeographical dynamics. The analytical aspects of the manuscript are especially strong. The paper will be of interest to a broad set of people who study avian biogeography; the dynamics of avian island community composition; the biogeography of the Indo-Pacific region; evolutionary biology of phylogeny and speciation processes; and conservation biologists who study persistence and extinction.

Because the study has been conducted in a thorough fashion, and the paper is well organized and is clearly written, my suggestions are relatively few and fairly minor, and mostly involve expanding some issues already noted. In no particular order, they are the following.

RESPONSE:

Thanks very much to the reviewer for this extremely encouraging appraisal.

1. The keywords seem sparse. Consider adding some of these: “endemism, extinction, persistence, speciation, taxon cycles, Wallacea, Bismarcks, Solomons”.

RESPONSE:

We have added endemism, speciation, and taxon cycles as keywords.

2. The term “Wallacea” has been used variably since being coined, and needs some definition here. When proposed by Merrill (1924) and described in greater detail by Dickerson et al. (1928), it had the Philippines as one of two core areas, the second being the southern region that the authors of this paper refer to as Wallacea. The authors’ usage is fairly common currently but is not universal, and some recent analyses have presented evidence supporting the original definition; for a summary of the long history of the term, see Ali and Heaney (2021, 2022). I suggest the authors make it explicit that they are using the term for what is mostly the Indonesian portion of Wallacea, following the primary research paper of their choice. Using a popular-level field guide to birds for their use of the term is probably not the best idea. To be clear, I don’t object to their decision to limit their study in the way they have; they just need to be more careful with their use of terminology.

RESPONSE:

We have edited lines 559–560, which now read: “We follow the conventional modern geographic delimitation of Wallacea as defined by Darlington (1957) (see Ali & Heaney [2021]).” We previously cited another field guide (Dutson, 2011) for the regional delimitation of the Bismarck and Solomon archipelagos, but instead we now cite Mayr & Diamond (2001), which is a more authoritative academic work (lines 561–562).

3. Similarly, the term “taxon cycles” has been used principally in two somewhat different ways. The authors point this out well into the paper (line 338 and following), but do so rather briefly. Since this model is an important part of their paper, it seems they should at least briefly note in the Introduction (lines 76-78) that they follow the definition and description of Ricklefs and Cox (1978) and Ricklefs and Bermingham (2002), rather than the original formulation by Wilson (1961). The primary difference is that Wilson emphasized that the pattern of colonization-speciation-competition-upslope migration-extinction was repeated many times, and that it is the repetition that gives rise to the notion of a cycle. Ricklefs mostly dropped the repetitiveness of the complete cycle, and emphasized repeated colonization, speciation, and upslope movement. What the authors have found in this study is indeed closest to what Ricklefs described, so any changes simply need to clarify what they mean (and do not mean) by “taxon cycle”.

RESPONSE:

Our study does de-emphasize the idea of “taxon cycles within taxon cycles” or “recycling,” and we agree that this is worth pointing out. We are concerned, though, that this distinction is too nuanced to discuss in the Introduction, and might be distracting. Instead, we address this in the Results section where we specifically address evidence for and against taxon cycles in light of modern distribution patterns (lines 223–227): “Note that although we here treat a taxon cycle as a process leading to a definite end point (extinction of relictual montane species), some conceptions of the taxon cycle also allow for the initiation of new cycles within in-progress cycles. This could complicate prediction of distributional patterns, but presumably any new lowland expansion would result in a species’ LIPs outnumbering its MIPs.”

4. A majority of the studies that are relevant to this paper have focused on birds, and so it is

appropriate that the authors emphasize that literature. However, they do so rather exclusively, without noting that their summaries of the literature emphasize birds. I suggest that they insert the word “avian” or “birds” appropriately to indicate this. I suggest that they also cite at least a few of the papers on other taxa (mammals, herps, insects) that are relevant to this study. It may be the case that few of these papers include “taxon cycle” in their titles, but that does not mean that they are not relevant.

RESPONSE:

We have added the word “bird” to line 79 in the Introduction to highlight that the four works cited in the following sentence (lines 80–81) all pertain to birds. In lines 85–88, we cited a series of papers supporting or refuting taxon cycles. We have now amended this series of references, removing two bird papers (Pepke et al., 2019 and Kennedy et al., 2022), and substituting a paper on plants (Hughes & Eastwood, 2006), and a paper on insects (Economo & Sarnat, 2012). The Introduction unavoidably cites many bird studies, but now includes citations to four works on insects, four on plants, and two encompassing multiple organism groups.

5. The authors show clear evidence that species of birds that have colonized montane areas widely within Wallacea (as they define it) are related to continental species that are migratory, and they view migratory behavior as an important pre-adaptation. That is clearly demonstrated by their data. However, they briefly touch on another aspect that they reject without due consideration, physiological adaptation to the cool climates that characterize montane habitats in the tropics. A knowledgeable colleague tells me that many of the birds on the Eurasian continent that are migratory have been studied and demonstrate physiological features that allow them to deal with cool/cold conditions while also retaining the ability to deal with hot conditions. This suggests to me that physiological plasticity and migratory behavior may both contribute to the success of these colonizing birds, and may even be parts of a tightly co-evolved set of adaptations. Currently, this hypothesis is given short shrift.

RESPONSE:

We don’t actually raise (or explicitly reject) this idea that migratory species have greater physiological plasticity and are therefore better colonizers. Relating back to the comment made by Reviewer 1 (regarding line 90), it does not seem safe to assume that migratory species have greater physiological tolerances; migrants may track their preferred climate niche, whereas residents must tolerate local seasonal changes.

6. A bit further afield is a topic that might deserve explicit mention. The calibrated phylogenies presented here show that many of these taxa evolved in the early Pleistocene, and some earlier. This means that they have persisted through many (perhaps as many as 24) major climatic (and sea level) fluctuations. This level of persistence seems notable in its own right; it suggests that persistence is an important part of this system, and that extinction can have had less than overwhelming effects; stated another way, turnover is less apparent than persistence. That topic may deserve some development.

RESPONSE:

This is an interesting idea. We are hesitant to tackle this subject, though, because we have very little information about passerine extinctions on these islands prior to their colonization by humans, and we aren’t equipped to make any quantitative comparisons regarding the roles of extinction vs. persistence.

Reviewer #3 (Remarks to the Author):

Dear Dr. Ward,

In the article "The formation of the Indo-Pacific montane avifauna" the authors try to understand the processes behind the formation of the avian diversity in mountain ranges in islands in Southeast Asia and part of Oceania. They use phylogenetic trees and ancestral range reconstruction to access the origin of 80 bird species complexes spread across their focal areas in Wallacea, Birmarcks and Solomons. In general, the article is well written, as the authors do a good job at contextualizing their goals and achievements in the existing theoretical framework. However, I have some concerns that should be addressed before I can recommend this paper for acceptance at Nature Communications:

RESPONSE:

Thanks very much to the reviewer for this positive appraisal, and for the constructive comments, which we have done our best to address.

1) The article's main flaws are in the dataset used to build most of their phylogenetic trees, which are a handful of nuclear and mitochondrial genes. Especially for applying BioGeoBEARS, I would expect phylogenetic trees built from reduced representation genomic data at the species/subspecies level. I am not proposing the authors go back and re-do their entire work, but they need to acknowledge the fact that their trees are just a hypothesis of branching patterns and so are the conclusions regarding ancestral ranges. It is also important to take in consideration the magnitude in which trees produced from mitochondrial markers are discordant in relation to trees from nuclear markers.

RESPONSE:

We see a broad taxonomic scope and dense species sampling (800 ingroup tips across 18 passerine families) as both a prerequisite and a major strength of this study, and it was not feasible to achieve anywhere near this level of sampling with reduced representation genomic data. We did, however, generate new genomic data for 61 individuals for this project for the detailed population studies of the most successful colonizers of island mountains. We certainly recognize that analyzing high numbers of nuclear loci can produce more robust evolutionary hypotheses, and that mitochondrial and nuclear markers can produce discordant signals. This is addressed directly in the revised version of the Methods section (lines 601–606): “Recent studies have demonstrated varying levels of discordance between phylogenetic analyses driven largely by mitochondrial genes versus those using dense sampling of nuclear markers (e.g., Andersen et al., 2019). Is therefore necessary to recognize a degree of topological uncertainty in the analyses described in this section, even in highly-supported relationships. However, this should not affect tree topologies so as to bias the ancestral state reconstructions and downstream analyses in any particular direction.”

We definitely did not intend to give the impression that our phylogenetic trees are “the truth” rather than hypotheses. We have edited the first paragraph of the Results (lines 129–130) as follows: “The ~~trees~~ phylogenetic hypotheses generated ~~produced~~ for this study were highly congruent with the latest published molecular analyses for the relevant groups.” Reviewer #3 repeats the concern expressed above that we treat the ancestral state reconstructions as “the true past” in their third comment; we address the issue under that point.

2) The paper has two major analytical approaches, one at the species level and one at the populational level for a few focal groups. Although the first part is framed in a hypothesis testing structure, the second part seems disconnected and needs to be better integrated with the paper main goal. There should be more information on how the focal species were selected and how their selection could bias the paper conclusions in one specific direction.

RESPONSE:

Please see our above response to Reviewer #1, who made a similar observation (comment on line 221 in the previous manuscript version). We agree that the presentation of the population studies was lacking in the previous manuscript version. To address this, we have shifted important information about the hypotheses and background from the Methods to the Results, and heavily edited this section to clarify the questions being asked and how they relate to the core aims of the study. The relevant section in the Results now begins with two paragraphs that adequately introduce and contextualize the population studies (lines 322–347).

As part of the expanded Results section, we have added an explanation of the selection criteria for the focal species/clades (lines 330–334): “They were chosen because they are among the most successful colonizers of island mountains in the focal region, and because it was possible to confidently infer their continental source pools — available evidence (Alström et al., 2011, 2018; Moyle et al., 2015) suggests Eurasian ancestral origins for all three.” Given these criteria, we did not analyze e.g. *Erythrura trichroa* (12 MIPs in the focal islands) because it doesn’t clearly trace back to Eurasia or Australo-Papua. *Turdus poliocephalus* (Eurasian, 10 MIPs) is the focus of a separate study (Reeve et al., 2023), and it was not feasible to obtain comprehensive geographic sampling for *Ficedula westermanni* (Eurasian, 9 MIPs). We are not sure what specific bias could be introduced by our selection of species. We make no corresponding population studies of Australo-Papuan-origin taxa, but there are few of these that have suitably large numbers of island populations, and the most obvious candidate, *Pachycephala orioloides* (Australo-Papuan, 1 MIP, 21+ LIPs) has been the focus of previous studies. We have added a note to that effect on lines 334–336: “Population studies of Australo-Papuan-origin passerines with broad island distributions but few MIPs can be found in Andersen et al. (2014) and Jönsson et al. (2014).”

3) In general, the language of the paper treats ancestral reconstructions as the "true past". The article language should reflect the fact that ancestral reconstructions are only hypotheses of past range and behaviors.

RESPONSE:

We have edited relevant passages to underscore to readers that the ancestral state reconstructions are inferences or hypotheses, not definitive statements about “the true past.”

In the Results, the second paragraph (lines 134–138) now begins, “We defined the geographic origin of the 80 focal species using ancestral ranges inferred with BioGeoBEARS (Matzke 2013a, 2013b). We counted back from terminal tree nodes until reaching “Ancestral Source Nodes” with > 75% probability of being either Eurasian (Palearctic + Indomalaya) or Australo-Papuan (see Methods: Ancestral state reconstructions).”

In Results (lines 150–152): “Eurasian-origin species generally trace back to ancestors with reconstructed distributions spanning both Indomalaya and the Palearctic (Supplementary Data 4).”

In Results: Migration (lines 244–246): “More than half of the ancestors of the 31 Eurasian-origin species ~~had~~ were inferred to have short-distance migrant populations, and very few had long-distance migrant populations (Supplementary Fig. 3; Table 2).”

In the Discussion (lines 389–391): “However, our ancestral geographic and elevational range reconstructions ~~show~~ indicate that virtually none of the montane diversity in the focal islands derives from lowland Indomalaya (Fig. 2; Supplementary Data 3).”

In Methods: Montane vs. lowland ancestry (lines 695–703): “To ~~establish~~ infer whether Eurasian-origin and Australo-Papuan-origin species evolved from lowland or montane continental ancestors, and whether there was a difference between the two groups, we used the reconstructed probabilities of montane distribution for the Ancestral Source Nodes of each species.”

4) The authors need to be especially careful with their results on competitive pressures, as they are using the shortcut of presence/absence of lineages in the same genus/family as evidence for competitive pressures between bird species. This a fragile assumption, especially considering that outcompeting can lead to extinction, and the methods applied here are practically blind to extinct lineages.

RESPONSE:

We understand this concern, especially in light of unclear and incomplete formulation in the previous version (see also Reviewer 1’s comment on lines 193-194 in the previous version). The relevant passage in the Results section (lines 268–274) now reads: “Competitive pressure from closely related lowland species is often implicated in the formation of new montane populations from lowland source pools (Terborgh, 1971). This mechanism may not be important for species colonizing directly between mountains and retaining their elevational niche. In line with these hypotheses, Australo-Papuan-origin MIPs often share islands with lowland relatives while Eurasian-origin MIPs do not. To quantify this pattern, we determined whether individual MIPs share islands with breeding species from the same genus and family (respectively) occurring in the lowlands.”

We have also added the following important caveat to the end of the relevant paragraph in the Results (lines 284–286): “We stress that the results presented in this section indicate the *potential* for competition from lowland relatives. Direct evidence requires field experiments (Jankowski et al., 2010; Freeman et al., 2016).”

We want to show that Eurasian-origin MIPs don’t maintain montane distributions due to competitive pressure from lowland relatives, and, because lowland relatives are mostly absent from the relevant islands, that seems like a safe conclusion regardless of possible past extinctions. It also seems unlikely that montane populations would outcompete closely related lowland species into extinction but then fail to expand their ranges into the lowlands. We think, therefore, that with due caution in interpretation, the results presented in this section are valid and worth including.

a few specific comments:

line 178: weird use of "overwhelmingly", probably replace with "mostly"

RESPONSE:

We have replaced “overwhelmingly” with “almost entirely” (line 248) to underscore that nearly all (mean = 99%) of ancestors were inferred to be entirely sedentary.

line 204-205: please explain what F being greater in empirical datasets than in simulated data means in the context of your hypothesis testing.

RESPONSE:

We have expanded the relevant paragraph (lines 289–307) to better explain this:

“We used phylogenetic null models to assess how well the empirical patterns of the above statistical tests could be replicated based on data simulated under a Brownian motion model of evolution (Supplementary Table 5). These tests include comparisons of Eurasian- versus Australo-Papuan-origin species: montane versus lowland ancestry, number of MIPs (both total and for individual regions), proportion of MIPs total island populations, and migration ancestry. We also tested, for Eurasian-origin species, number of MIPs for short-distance migrant versus sedentary species, and number of MIPs for species with continental populations vs. those restricted entirely to islands. The phylogenetic null models were performed to determine how well our results are explained by phylogenetic history alone, in the absence of direct quantification of geographic or ecological characters among lineages. For most of the aforementioned analyses, our empirical F statistics are significantly greater than those derived from the simulated null datasets. The F statistic represents the ratio of the variance between and within groups; the value becomes higher as the variance between groups increases relative to the variance within them. When the empirical value of F is greater than the distribution of values simulated under Brownian motion, we conclude that the patterns in these particular variables are greater than would be predicted based purely on phylogeny. However, the particular pattern that Eurasian-origin species have more MIPs than Australo-Papuan-origin species can be reproduced upon simulating the data using Brownian motion alone. This is true with regard to all islands ($p = .07$), for Wallacea alone ($p = .16$), and for Bismarcks/Solomons alone ($p = .08$).”

line 274: please rephrase acknowledging the limitations of your results.

RESPONSE:

We have revised this passage to sound less unequivocal, and to tie statements to particular results/figures (lines 403–411): “Montane island populations experience different fates depending on their ancestral origins. For Eurasian-origin species, mountains often represent stepping stones to additional montane colonizations. Our analyses indicate that they retain an ancestral montane niche as they disperse, often extensively, across archipelagos (Figs. 2–4; Supplementary Figs. 4–15). Conversely, Australo-Papuan-origin species appear to have arisen via past island colonizations by lowland ancestors (Fig. 2), and little in their modern distribution patterns indicates an ability to colonize directly between the mountains of different islands (Figs. 3 and 4; see also Results: MIPs and LIPs). For these species, mountains represent dead ends for further dispersal.”

Line 624: this section needs to be better explained. How was Oliveros’ tree reconciled with your trees?

RESPONSE:

We have expanded this section to better explain the methods we used (lines 791–797): “We used a well-resolved, dated ultraconserved element (UCE) phylogenetic tree of passerine families (Oliveros et al., 2019) as a backbone to represent the interrelationships among groups. We first pruned the tips of their consensus tree so that only the focal clades represented in our study remained. Next, we pruned down the clade-level phylogenies generated for our own study to include only species with MIPs (one individual per species). We then grafted these trees onto the Oliveros et al. backbone.”

REFERENCES

- Ali, J. R., & Heaney, L. R. (2021). Wallace's line, Wallacea, and associated divides and areas: history of a tortuous tangle of ideas and labels. *Biological Reviews*, 96(3), 922-942.
- Alström, P., Höhna, S., Gelang, M., Ericson, P. G., & Olsson, U. (2011). Non-monophyly and intricate morphological evolution within the avian family Cettiidae revealed by multilocus analysis of a taxonomically densely sampled dataset. *BMC Evolutionary Biology*, 11(1), 352.
- Alström, P., Rheindt, F. E., Zhang, R., Zhao, M., Wang, J., Zhu, X., ... & Prawiradilaga, D. M. (2018). Complete species-level phylogeny of the leaf warbler (Aves: Phylloscopidae) radiation. *Molecular Phylogenetics and Evolution*, 126, 141–152.
- Andersen, M. J., McCullough, J. M., Friedman, N. R., Peterson, A. T., Moyle, R. G., Joseph, L., & Nyari, A. S. (2019). Ultraconserved elements resolve genus-level relationships in a major Australasian bird radiation (Aves: Meliphagidae). *Emu-Austral Ornithology*, 119(3), 218-232.
- Andersen, M. J., Nyári, Á. S., Mason, I., Joseph, L., Dumbacher, J. P., Filardi, C. E., & Moyle, R. G. (2014). Molecular systematics of the world's most polytypic bird: the *Pachycephala pectoralis/melanura* (Aves: Pachycephalidae) species complex. *Zoological Journal of the Linnean Society*, 170(3), 566-588.
- Chapman, F. M. (1917). The distribution of bird-life in Colombia. *Bulletin of the AMNH*, 36:1-729.
- Chapman, F. M. (1926). The distribution of bird-life in Ecuador. *Bulletin of the AMNH*, 55:1-784
- Darlington, P. J. (1957). *Zoogeography: The Geographical Distribution of Animals*. John Wiley & Sons, New York.
- Donoghue, M. J. (2008). A phylogenetic perspective on the distribution of plant diversity. *Proceedings of the National Academy of Sciences*, 105(supplement_1), 11549–11555.
- Dutson, G. (2011). *Birds of Melanesia: Bismarcks, Solomons, Vanuatu and New Caledonia*. Christopher Helm, London.

- Economu, E. P., & Sarnat, E. M. (2012). Revisiting the ants of Melanesia and the taxon cycle: historical and human-mediated invasions of a tropical archipelago. *The American Naturalist*, *180*(1), E1-E16.
- Freeman, B. G., Class Freeman, A. M., & Hochachka, W. M. (2016). Asymmetric interspecific aggression in New Guinean songbirds that replace one another along an elevational gradient. *Ibis*, *158*(4), 726–737.
- Gómez, C., Tenorio, E. A., Montoya, P., & Cadena, C. D. (2016). Niche-tracking migrants and niche-switching residents: evolution of climatic niches in New World warblers (Parulidae). *Proceedings of the Royal Society B: Biological Sciences*, *283*(1824), 20152458.
- Hughes, C., & Eastwood, R. (2006). Island radiation on a continental scale: exceptional rates of plant diversification after uplift of the Andes. *Proceedings of the National Academy of Sciences*, *103*(27), 10334-10339.
- Jankowski, J. E., Robinson, S. K., & Levey, D. J. (2010). Squeezed at the top: Interspecific aggression may constrain elevational ranges in tropical birds. *Ecology*, *91*(7), 1877–1884.
- Jönsson, K. A., Irestedt, M., Christidis, L., Clegg, S. M., Holt, B. G., & Fjeldså, J. (2014). Evidence of taxon cycles in an Indo-Pacific passerine bird radiation (Aves: *Pachycephala*). *Proceedings of the Royal Society B: Biological Sciences*, *281*(1777), 20131727.
- Kennedy, J. D., Marki, P. Z., Reeve, A. H., Blom, M. P., Prawiradilaga, D. M., Haryoko, T., ... & Jönsson, K. A. (2022). Diversification and community assembly of the world's largest tropical island. *Global Ecology and Biogeography*, *31*(6), 1078–1089.
- Laube, I., Graham, C. H., & Böhning-Gaese, K. (2015). Niche availability in space and time: migration in *Sylvia* warblers. *Journal of Biogeography*, *42*(10), 1896-1906.
- Linck, E., Freeman, B. G., & Dumbacher, J. P. (2020). Speciation and gene flow across an elevational gradient in New Guinea kingfishers. *Journal of Evolutionary Biology*, *33*(11), 1643-1652.
- Lobo, J. M., & Halffter, G. (2000). Biogeographical and ecological factors affecting the altitudinal variation of mountainous communities of coprophagous beetles (Coleoptera: Scarabaeoidea): a comparative study. *Annals of the Entomological Society of America*, *93*(1), 115–126.
- Matzke, N. (2013a). Probabilistic historical biogeography: new models for founder-event speciation, imperfect detection, and fossils allow improved accuracy and model-testing. University of California, Berkeley.
- Matzke, N.J. (2013b). BioGeoBEARS: Biogeography with Bayesian (and Likelihood) Evolutionary Analysis in R Scripts. University of California, Berkeley.

- Mayr, E., & Diamond, J. M. (2001). *The Birds of Northern Melanesia: Speciation, Ecology & Biogeography*. Oxford University Press, New York.
- Merckx, V. S., Hendriks, K. P., Beentjes, K. K., Mennes, C. B., Becking, L. E., Peijnenburg, K. T., ... & Schilthuizen, M. (2015). Evolution of endemism on a young tropical mountain. *Nature*, *524*(7565), 347–350.
- Moyle, R. G., Hosner, P. A., Jones, A. W., & Outlaw, D. C. (2015). Phylogeny and biogeography of *Ficedula* flycatchers (Aves: Muscicapidae): novel results from fresh source material. *Molecular Phylogenetics and Evolution*, *82*, 87–94.
- Moyle, R. G., Manthey, J. D., Hosner, P. A., Rahman, M., Lakim, M., & Sheldon, F. H. (2017). A genome-wide assessment of stages of elevational parapatry in Bornean passerine birds reveals no introgression: implications for processes and patterns of speciation. *PeerJ*, *5*, e3335.
- Oliveros, C. H., Field, D. J., Ksepka, D. T., Barker, F. K., Aleixo, A., Andersen, M. J., ... & Faircloth, B. C. (2019). Earth history and the passerine superradiation. *Proceedings of the National Academy of Sciences*, *116*(16), 7916-7925.
- Pepke, M. L., Irestedt, M., Fjeldså, J., Rahbek, C., & Jønsson, K. A. (2019). Reconciling supertramps, great speciators and relict species with the taxon cycle stages of a large island radiation (Aves: Campephagidae). *Journal of Biogeography*, *46*(6), 1214–1225.
- Pujolar, J. M., Blom, M. P., Reeve, A. H., Kennedy, J. D., Marki, P. Z., Korneliussen, T. S., Freeman, B. G., Sam, K., Linck, E., Haryoko, T., Iova, B., Koane, B., Maiah, G., Paul, L., Irestedt, M., & Jønsson, K. A. (2022). The formation of avian montane diversity across barriers and along elevational gradients. *Nature Communications*, *13*(1), 1–13.
- Reeve, A. H., Gower, G., Pujolar, J. M., Smith, B. T., Petersen, B., Olsson, U., ... & Jønsson, K. A. (2023). Population genomics of the island thrush elucidates one of earth's great archipelagic radiations. *Evolution Letters*, *7*(1), 24-36.
- Sheldon, F. H., Lim, H. C., & Moyle, R. G. (2015). Return to the Malay Archipelago: the biogeography of Sundaic rainforest birds. *Journal of Ornithology*, *156*(1), 91–113.
- Terborgh, J. (1971). Distribution on environmental gradients: theory and a preliminary interpretation of distributional patterns in the avifauna of the Cordillera Vilcabamba, Peru. *Ecology*, *52*(1), 23–40.

Reviewers' Comments:

Reviewer #2:

Remarks to the Author:

The authors have responded suitably to my earlier comments and recommendations. I have nothing further to suggest.

Reviewer #3:

Remarks to the Author:

Thank you for the explanations and for addressing my concerns and suggestions. I believe the new manuscript has improved and it is a solid and broadly interesting article.

REVIEWERS' COMMENTS

Reviewer #2 (Remarks to the Author):

The authors have responded suitably to my earlier comments and recommendations. I have nothing further to suggest.

RESPONSE: We are very pleased that the reviewer is satisfied with the revised manuscript, and are grateful for the constructive criticism provided.

Reviewer #3 (Remarks to the Author):

Thank you for the explanations and for addressing my concerns and suggestions. I believe the new manuscript has improved and it is a solid and broadly interesting article.

RESPONSE: We are very pleased that the reviewer is satisfied with the revised manuscript, and are grateful for the constructive criticism provided.